# Perpendicular electric field drives Chern transitions and layer polarization changes in Hofstadter bands

Pratap Chandra Adak [1] ✉, Subhajit Sinha [1], Debasmita Giri[2], Dibya Kanti Mukherjee[3,4,5], Chandan[1], L. D. Varma Sangani[1], Surat Layek [1], Ayshi Mukherjee[1], Kenji Watanabe [6], Takashi Taniguchi [7], H. A. Fertig[3,4], Arijit Kundu [2] ✉ & Mandar M. Deshmukh [1] ✉

Moiré superlattices engineer band properties and enable observation of fractal energy spectra of Hofstadter butterfly. Recently, correlated-electron physics hosted by flat bands in small-angle moiré systems has been at the foreground. However, the implications of moiré band topology within the single-particle framework are little explored experimentally. An outstanding problem is understanding the effect of band topology on Hofstadter physics, which does not require electron correlations. Our work experimentally studies Chern state switching in the Hofstadter regime using twisted double bilayer graphene (TDBG), which offers electric field tunable topological bands, unlike twisted bilayer graphene. Here we show that the nontrivial topology reflects in the Hofstadter spectra, in particular, by displaying a cascade of Hofstadter gaps that switch their Chern numbers sequentially while varying the perpendicular electric field. Our experiments together with theoretical calculations suggest a crucial role of charge polarization changing concomitantly with topological transitions in this system. Layer polarization is likely to play an important role in the topological states in few-layer twisted systems. Moreover, our work establishes TDBG as a novel Hofstadter platform with nontrivial magneto-electric coupling.

The 2D moiré lattice, when subjected to a magnetic field, loses its periodicity due to spatial dependence of the gauge potential. However, when the applied magnetic field is such that the magnetic flux quantum per unit cell of the moiré lattice is a rational number, the discrete translational symmetry of the lattice is restored with a larger magnetic unit cell. The energy spectrum of such a system, as a function of the magnetic field, has a self-similar fractal structure known as Hofstadter's butterfly[1]. The observation of this quantum fractal is limited by the requirement of high magnetic flux $\Phi$ through the unit cell, such that $\Phi/\Phi_0 \sim 1$. Here, $\Phi_0 = h/e$ is the magnetic flux quantum, with $h$ being Planck's constant and $e$ being the electron charge. Hofstadter's butterfly was first observed in graphene aligned to hexagonal boron nitride (hBN)[2,3]. The large unit cell in such moiré superlattices realizes $\Phi/\Phi_0 \sim 1$ with available lab magnets.

[1]Department of Condensed Matter Physics and Materials Science, Tata Institute of Fundamental Research, Homi Bhabha Road, Mumbai 400005, India. [2]Department of Physics, Indian Institute of Technology, Kanpur 208016, India. [3]Department of Physics, Indiana University, Bloomington, IN 47405, USA. [4]Quantum Science and Engineering Center, Indiana University, Bloomington, IN 47408, USA. [5]Laboratoire de Physique des Solides, Univ. Paris-Sud, Université Paris Saclay, CNRS, UMR 8502, F-91405 Orsay Cedex, France. [6]Research Center for Functional Materials, National Institute for Materials Science, 1-1 Namiki, Tsukuba 305-0044, Japan. [7]International Center for Materials Nanoarchitectonics, National Institute for Materials Science, 1-1 Namiki, Tsukuba 305-0044, Japan. ✉e-mail: pratapchandraadak@gmail.com; kundua@iitk.ac.in; deshmukh@tifr.res.in

Recently, the ability to stack multiple layers of 2D materials rotated with sub-degree precision has opened up a new frontier. In addition to tuning the moiré length scale, the twist angle between two adjacent layers tunes the symmetry and the topology of the emergent moiré bands, providing new experimental knobs. Furthermore, magic-angle twisted bilayer graphene (TBG) hosts low-energy flat bands[4–6] that support correlated-electron phenomena such as correlated insulator states[7–9], ferromagnetism[10,11], and superconductivity[9,12]. Topological properties of the twisted systems are of particular interest, as several recent studies have explored correlated Chern insulator states in the Hofstadter regime in TBG[13–18]. These states are interpreted as arising due to the occupation of subsets of underlying Chern bands[13–15] or Hofstadter subbands[16–18], a mechanism similar to Quantum Hall ferromagnetism. In the physics of Chern insulator states, as also in the quantum Hall physics due to the formation of Landau levels, gaps with different Chern numbers can be accessed by changing the Fermi energy by varying the charge density. Recently a pure electrical control, such as the perpendicular electric field, to open up a Chern insulating state from a bulk gapless state has been demonstrated using a correlated system[19]. Similar electrical control over Chern states without requiring electron correlation will be novel.

Twisted double bilayer graphene (TDBG), made by twisting two copies of Bernal stacked bilayer graphene (BLG), provides such opportunities as the electric field can tune the band structure and its topological properties[20–26]. Notably, the flat bands in TDBG possess a nonzero valley Chern number that changes with the electric field, offering a unique correlated Hofstadter platform[27]. While earlier experiments in TDBG have a major focus on electron correlations physics[28–34], the tunability of the topological flat bands in the Hofstadter regime is little explored[35]. An ability to tune the Chern numbers of these bands would provide further insight on the role of topology in these correlated states. Thus TDBG is a rich platform as the electric field plays an important role, unlike in TBG.

In this work, we study electron transport in TDBG with small twist angles around $1.1°–1.5°$ under a high magnetic field upto $\Phi/\Phi_0 \sim 1/3$. We observe a cascade of gaps that change their Chern numbers sequentially as the perpendicular electric field is varied. This contrasts with TBG for which the band structure is unaffected by such an electric field, and so cannot induce Chern transition. The Hofstadter fan diagrams we measure show additional features that reveal the topological nature of the underlying band structure. Our calculation of the Hofstadter energy spectrum in TDBG confirms the key experimental observations. Interestingly, we find that a small exchange enhanced spin Zeeman term plays a role in determining the sequence of Chern gaps. Furthermore, our analysis shows that the electric field varies the layer polarization and provides the underlying mechanism of the Chern transition.

## Results

### Low-temperature transport

We now present our magneto-transport measurements in TDBG devices. To fabricate TDBG devices, we cut two pieces of BLG from a single exfoliated flake and sandwich them between two hBN flakes with a relative rotation[36]. A schematic of our device structure is shown in Fig. 1a. Using the metal top gate and Si$^{++}$ bottom gate we can independently control the charge density, $n$, and the perpendicular electric displacement field, $D$. Using the multiple electrodes in the devices we measure low-temperature electron transport in a Hall bar geometry under a perpendicular magnetic field, $B$. See Methods section for details about fabrication and measurement.

We first discuss electron transport in zero magnetic field. In Fig. 1b, we calculate the zero magnetic field band structures of TDBG with a twist angle of $1.10°$ for two different electric fields (see Supplementary Note 1 for calculation details). In addition to the tunability of the band gaps and width, the valley Chern numbers of the moiré bands

change with the electric field. In Fig. 1c, we show a color-scale plot of the longitudinal conductivity, $\sigma_{xx}$, as a function of $v$ and $D$ at $B = 0$ T and a temperature of 300 mK for a TDBG device with twist angle $1.10°$. Here, $v = 4n/n_S$ is the moiré filling factor, where $n_S$ is the number of charge carriers required to fill an isolated moiré band and the factor 4 incorporates the spin and valley degeneracy. In the color-scale plot of $\sigma_{xx}$, conductivity dips are observed corresponding to two moiré gaps at $v = \pm 4$ and the CNP gap at $v = 0$. The electric field tunability of the underlying band structure is evident as two moiré gaps close at a high electric field, and the CNP gap opens up only above a finite electric field value. We also see additional regions with low conductance – a cross-like feature around $D/\epsilon_0 = 0$ and $v \sim -2$ on the hole side and two halo regions around $D/\epsilon_0 = \pm 0.3$ V/nm and $v \sim 2$ on the electron side. These are characteristic features of small-angle TDBG[37]. Here we note, while the correlated gaps at partial filling develop in TDBG with twist angle $\sim 1.2°–1.4°$, for smaller twist angle $\sim 1.1°$ correlated gaps develop under a parallel magnetic field[23,28,30]. See Supplementary Fig. 8 for data from a $1.46°$ device, where we observe correlated gap at a zero magnetic field.

### Observation of electric field tunable Chern gaps in high magnetic field

To study the effect of a perpendicular magnetic field $B$, we now measure $\sigma_{xx}$ as a function of $v$ and $D$ for different values of $B$ as shown in Fig. 1d–f. In contrast to the case of $B = 0$, the CNP gap emerges even at $D = 0$ as we apply a finite magnetic field (see Fig. 1d). Furthermore, as evident from the $\sigma_{xx}$ plot at 5 T in Fig. 1e, the CNP gap undergoes multiple closing and reopening as the electric field is varied. At higher magnetic fields, such as at 9 T in Fig. 1f, the formation of Landau Levels (LL) leads to multiple lines of conductivity peaks and dips parallel to the $D$-axis. Our most interesting observation is that the positions of the $\sigma_{xx}$ dips shift discretely on the $v$-axis as the electric field is varied. This effect of the electric field is most prominent in the region $|D|/\epsilon_0 \lesssim 0.25$ V/nm (indicated by the dashed rectangle in Fig. 1f). Interestingly, within this $D$-range, marked by the closing of the hole-side moiré gap at $|D|/\epsilon_0 \sim 0.25$ V/nm at zero magnetic field, both the flat bands are isolated from the remote moiré bands. At higher electric fields, as the flat bands merge with the remote moiré bands, the electric field tunability becomes weaker.

We now study the step-like evolution of the $\sigma_{xx}$ dips with $D$ in detail by focusing on $\sigma_{xx}$ as a function of $v$ and $D$ at $B = 9$T in Fig. 2a. To elucidate the nature of these dips as they evolve in the three-dimensional parameter space of $v$, $D$, and $B$, we perform a systematic analysis. We first identify two groups of most prominent $\sigma_{xx}$ dips corresponding to the larger gaps (see the Supplementary Note 3 and Supplementary Fig. 2 for estimation of the gaps), which shift further on the hole side as $D$ is increased, by line segments of different colors. Then to track the evolution of these dips with $B$, we measure $\sigma_{xx}$ as a function of $v$ and $B$ at different values of constant $D$. In Fig. 2b, we show three such plots, namely fan diagrams, for three different $D$. We focus on the pair of prominent $\sigma_{xx}$ dips in each fan diagram. The position of the dips on the $v$-axis evolve with $B$ along linear trajectories marked with lines of the same colors as in Fig. 2a. Besides the marked pair of dips, there are other dips which are also tuned by the electric field (Supplementary Fig. 4). See Supplementary Note 4 and Supplementary Figs. 7–9 for similar data from devices with different twist angles.

The linear trajectories of the conductivity dips can be understood in terms of Hofstadter physics. Within the Hofstadter picture, the gaps with integer Chern number $C$ follow linear trajectories in $n$-$B$ diagram, i.e., the Wannier diagram, given by the Diophantine equation,

$$v = C\frac{\Phi}{\Phi_0} + s. \tag{1}$$

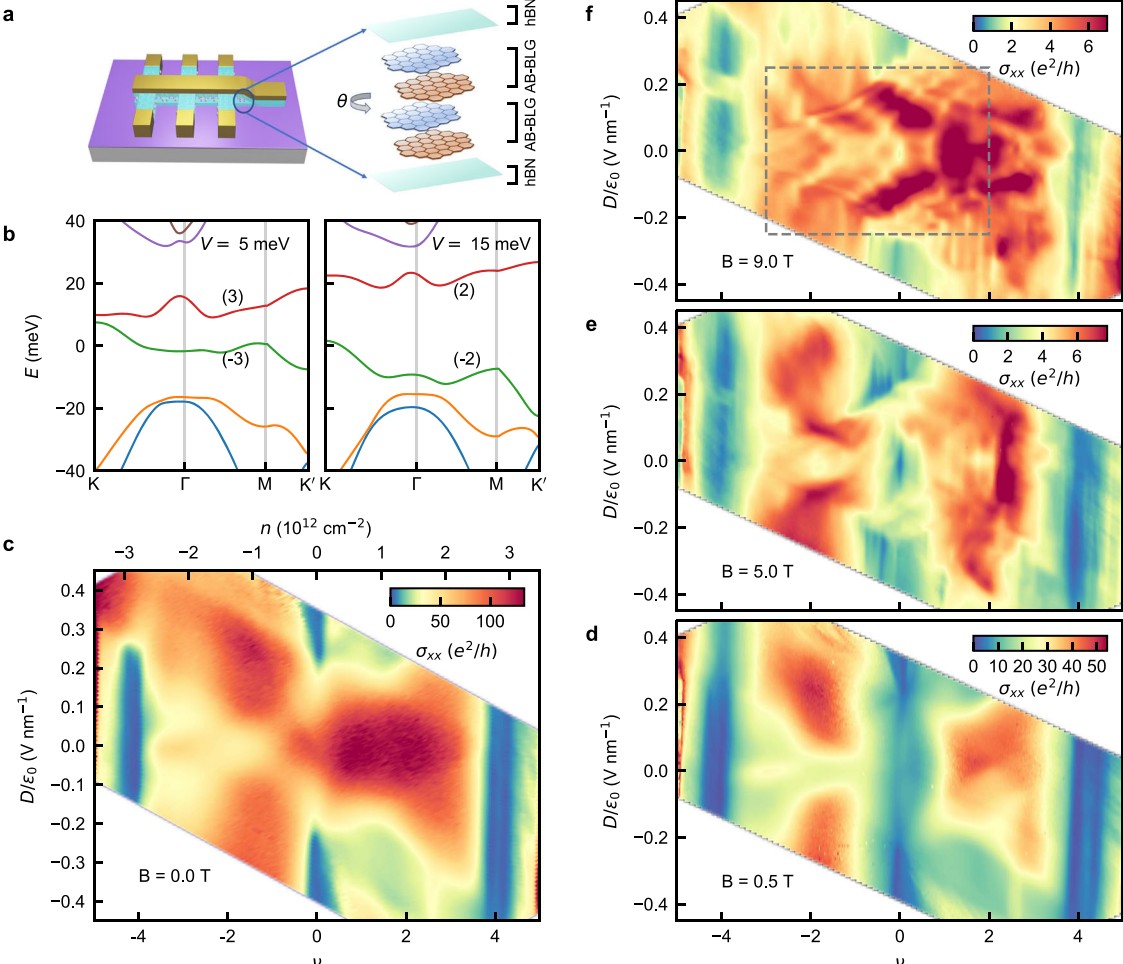

**Fig. 1 | Magneto-transport in twisted double bilayer graphene (TDBG).**
**a** Schematic of the dual gated TDBG device. **b** Calculated band structure for TDBG with twist angle 1.10° in absence of the magnetic field for two different values of the interlayer potential $V$. Interlayer potential simulates the application of the electric field. The numbers in brackets adjacent to each band indicate the corresponding valley Chern number, which changes with $V$. **c** $\sigma_{xx}$ as a function of moiré filling $\nu$ and perpendicular electric displacement field $D$ at zero magnetic field at 300 mK. **d–f** Variation of $\sigma_{xx}$ vs. $\nu$ and $D$ at three different values of perpendicular magnetic field, $B$. The region inside the dashed rectangle in **f** is discussed in details in Fig. 2.

Here, $s$ is an integer denoting the moiré filling factor corresponding to the number of carriers per moiré unit cell in zero magnetic field, and $\Phi = BA$, with $A$ being the area of the moiré unit cell. We extract $C$ and $s$ from the slope and $\nu$-axis intercept in the fan diagrams at different $D$ for all the marked lines. The extracted values of $(C, s)$ are marked in Fig. 2b. Then we assign these $(C, s)$ values inferred from Fig. 2b to the corresponding $\sigma_{xx}$ minima in Fig. 2a. Here we note that from the evolution of $\sigma_{xx}$ dips with temperature we extract the Hofstadter gaps to be ~0.1 – 0.5 meV (see Supplementary Note 3 and Supplementary Fig. 2). Our experimental data of $\sigma_{xy}$ (Supplementary Fig. 5) show weak quantization possibly due to the smallness of these gaps together with angle inhomogeneity disorder[38]. See Supplementary Note 6 and Supplementary Fig. 11 for the role of flat band energy scale in Hofstadter spectra. Also, moiré commensurability aspects can lead to absence of quantization[39].

From Fig. 2a, we find two interesting trends in the transition of $(C, s)$ as a function of $D$. Firstly, for both the groups corresponding to $s = 0$ and $s = -2$, the Chern number decreases sequentially by 1 as the magnitude of $D$ is increased; crucially, we observe both even and odd Chern numbers. Secondly, the difference in $C$ for $s = 0$ and $s = -2$ remains 2. The sequential change of Chern number by $D$ can be empirically understood by considering a simple schematic in Fig. 2c, where a part of the Chern band peels off due to varying electric field. As $D$ is varied, branches with Chern number 1 are separated from a group

of Hofstadter subbands and merge with another group, resulting in a sequence of Chern gaps with different $C$. A dip in $\sigma_{xx}$, observed experimentally, corresponds to a gap between two groups of Hofstadter subbands. This simple picture of the sequential evolution of the Chern gaps with electric field as Hofstadter subbands peel off is confirmed by our theoretical calculation which we discuss next.

## Calculation of Hofstadter spectra in TDBG
To calculate the Hofstadter energy spectrum in TDBG, we construct a Hamiltonian for each valley in the basis of bare Landau levels of graphene, indexed by the Landau level index, guiding center, and layer index. Inter-bilayer tunneling then couples states with various guiding centers and Landau levels of each layer and one can diagonalize the Hamiltonian to find the spectra. The prominent Chern gaps are characterized as a function of the filling factor $\nu$, calculated from the charge neutrality point and flux quanta per moiré unit cell, $\Phi/\Phi_0$, obtained by solving the Diophantine equation (Eq. (1)). In Fig. 3a, we plot the Hofstadter spectra for both $K$ and $K'$ valleys in two different colors for an interlayer potential of $V = 20$ meV. Here we note that at nonzero $V$ the valley degeneracy is already lifted due to the nontrivial topology of the underlying band structure, as we discuss later. The observation of odd Chern numbers further implies that the spin degeneracy is also lifted. To incorporate this in our calculation we include an exchange enhanced spin Zeeman term $\Delta_S = 2.5$ meV; such term can arise due to

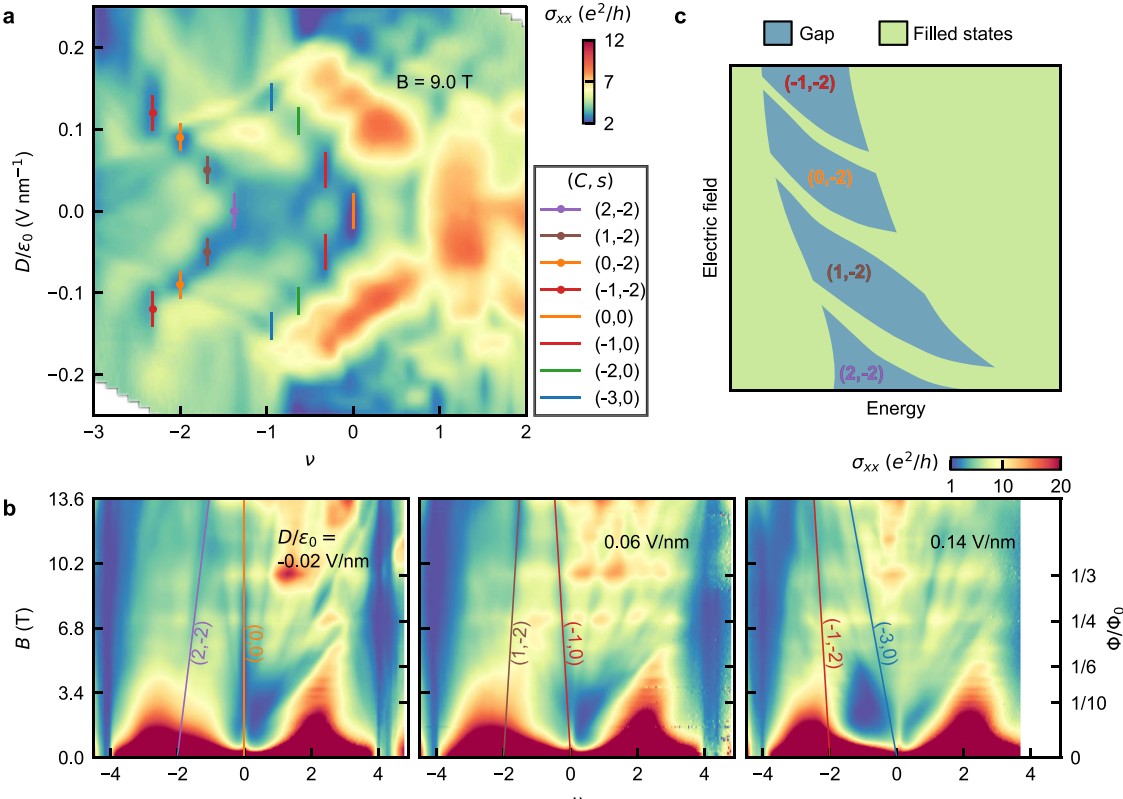

**Fig. 2 | Electric field tunable Chern gaps. a** Color-scale plot of $\sigma_{xx}$ as a function $\nu$ and $D$ at $B = 9$T. The overlayed lines of different colors indicate the $(C, s)$ values of the corresponding $\sigma_{xx}$ dips as in the legend. $(C, s)$ values are inferred from **b** with $C$ being the Chern number and $s$ being the moiré filling factor corresponding to $B = 0$ T (see Supplementary Fig. 6 for fitting details to extract $(C, s)$ values). **b** Color-scale plots of $\sigma_{xx}$ as a function of $\nu$ and $B$, i.e., fan diagrams, at three different values of $D$. The overlayed lines trace the trajectory of the $\sigma_{xx}$ minima. The slope and the $\nu$-axis intercept give $C$ and $s$, respectively. $(C, s)$ values are indicated adjacent to the lines and assigned in **a** accordingly. **c** Schematic representation of the origin of electric field tunable Chern gaps. Chern number of the gaps changes when Hofstadter subbands with Chern number 1 peel off from a group of Hofstadter subbands and merge with another group as the electric field is varied.

electron correlations of the flat bands. See Supplementary Note 2 for the details of the calculation. In Fig. 3b, we plot the Wannier diagram showing the evolution of the Chern gaps as a function of $\nu$ and $\Phi/\Phi_0$ for $V = 20$ meV. Here the width of the line segments indicates the strength of the corresponding gaps considering both $K$ and $K'$ valleys. See Supplementary Fig. 1 for calculation at other electric field values. Similar to our experimental observation, we find that two prominent Chern gaps originate from $s = 0$ and $s = -2$ and their Chern numbers change with the electric field.

To further elucidate the electric field-induced quantum phase transition of Chern numbers we plot the evolution of the Hofstadter energy spectrum as a function of the inter-layer potential $V$ at a constant $\Phi/\Phi_0 = 0.32 \cdot B = 9$T in Fig. 3c. We find that the energy levels disperse nonmonotonically with $V$. As a result, branches of a Chern band peel off and merge with another band, giving rise to gaps with different Chern numbers at different electric fields. The corresponding plot of extracted Chern gaps as a function $\nu$ and $V$ is shown in Fig. 3d. Interestingly, the sequence of changes in the prominent Chern gaps in Fig. 3d match quite well with our experimental data in Fig. 2a. Moreover, the Chern gaps are more prominent on the hole side, again as seen in experiment.

**Role of electric field-tunable layer polarization**

To understand the underlying mechanism further, we now discuss the role of an induced layer polarization in tuning the Hofstadter spectra. Under an applied electric field, electrons occupying the four different layers of TDBG are at different values of on-site potential, $V_i$, $i = 1, 2, 3, 4$. This contributes a mean-field energy of the form $E \sim \sum \rho_i V_i$, where $\rho_i$ is the electron density of $i$-th layer. As $V_i$ changes

proportionally with the electric field, the nonmonotonic change in $E$ with the electric field in Fig. 3c suggests an asymmetric change in the distribution of $\rho_i$. To measure the asymmetric change in the distribution, we define an energy band specific charge polarization across layers, $P = \rho_1 + \rho_2 - \rho_3 - \rho_4$. The role of this layer polarization in determining the energy vs. electric field dispersion is depicted schematically in Fig. 3e. For an electron distribution polarized toward the top layer, the energy increases with an increasing electric field, and hence the energy vs. electric field dispersion has a positive slope. The slope is negative for the opposite polarization. As the layer polarization is varied by the electric field, the energy state evolves nonmonotonically with the electric field. When two energy bands with different Chern numbers cross each other, the Chern number of the gap between the two bands changes. The electric field tunable layer polarization is indeed confirmed in Fig. 3f where we plot the polarization vs. $V$ for two different energy levels across the CNP. Figure 3e further suggests how the tunable layer polarization can manifest itself into a complex evolution of a gap, with the possibility of multiple closings and reopenings[40]. Indeed, we verify this interesting implication as we measure $\sigma_{xx}$ at the CNP gap as a function of $B$ and $D$. We find a complex evolution of the CNP gap with multiple closings and reopenings, a feature distinct from other materials like BLG, suggesting important role of field tunable layer polarization in twisted systems (details in the Supplementary Note 5 and Supplementary Fig. 10).

**Role of topology on Hofstadter spectra**

Finally, we discuss the important role of the nontrivial topology of the TDBG band structure in determining the Hofstadter spectra. At finite values of the electric field, the valley Chern numbers of $K$ and $K'$ valleys

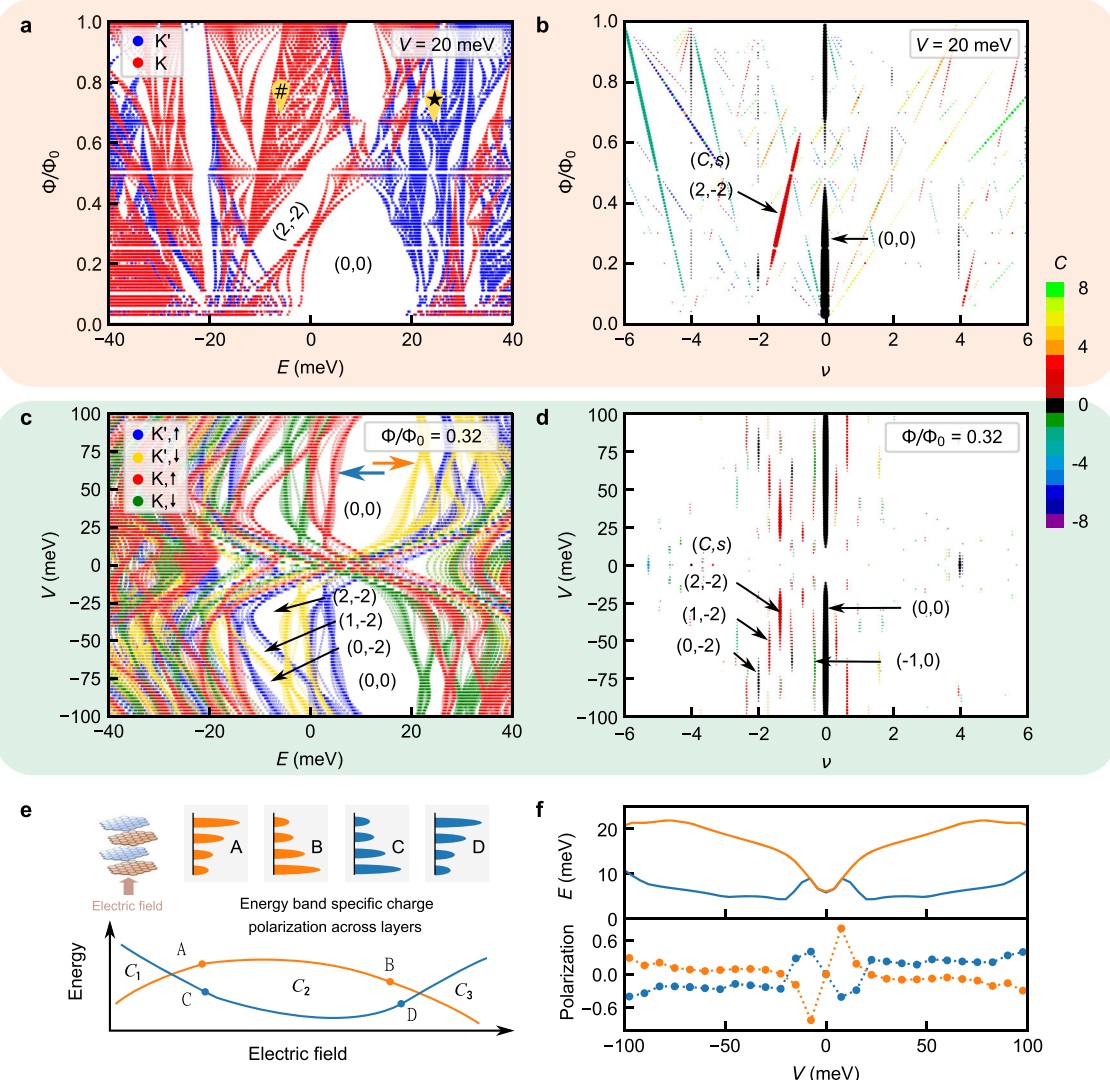

**Fig. 3 | Calculation of Hofstadter spectra and layer polarization in TDBG.**
**a** Calculated Hofstadter energy levels for both $K$ and $K'$ valleys in TDBG with twist angle 1.10° for an interlayer potential of $V = 20$ meV. Energy levels disperse differently for two valleys due to different topology resulting in fewer number of gaps-- e.g., energy gap for $K$ valley at location ★ is filled with energy levels from $K'$ valley (vice versa for location #). **b** Corresponding Wannier diagram showing the evolution of Hofstadter gaps considering both the valleys as a function $\nu$ and $\Phi/\Phi_0$. **c** Evolution of energy spectra as a function of interlayer potential $V$ at a constant magnetic field corresponding to $\Phi/\Phi_0 = 0.32$ - 9T with an exchange enhanced Zeeman splitting of 2.5 meV. This confirms the change in Chern numbers as schematically depicted in Fig. 2c. **d** Evolution of gaps as a function of $\nu$ and $V$ extracted from **c**. The width of the line segments in **b** and **d** are proportional to the values of

the gaps in the Hofstadter spectrum. The color of the line segments denotes the corresponding Chern number $C$, as indicated in the color-scale legend. **e** Schematic depicting the role of energy band-specific charge polarization across layers in energy vs. electric field dispersion. When electrons are polarized more toward top (bottom) layers, as for points A and D (B and C), the energy of the band increases (decreases) with increasing electric field. As the layer polarization is varied by the electric field, energy disperses nonmonotonically with the electric field inducing multiple gap closing and opening -- this can lead to changes in the Chern number of the gap. **f** Calculated layer polarization (bottom panel) corresponding to two Hofstadter bands (top panel) across the CNP gap (0,0) (indicated by two arrows in **c**). Panels **a** and **b** are shaded in the same color to indicate that they are extracted from the same data (similarly for panels **c** and **d**).

---

are nonzero and opposite in sign. As seen form Fig. 3a, the Hofstadter energy spectra from two valleys disperse differently with the magnetic field due to the different topologies of the two valleys[27]. This has two important implications. Firstly, the two-fold valley degeneracy is lifted, as we noted earlier. Secondly, only the most prominent gaps survive as gaps in the Hofstadter spectrum of one valley can be filled by the energy states of the other. This is evidenced in our experimental data as well.

We find additional signatures of the nontrivial band topology in the Hofstadter energy spectra in TDBG. The Hofstadter spectrum of a topologically trivial band is confined within the band, i.e., disconnected from the spectra of the neighboring bands. Conversely, the Hofstadter spectrum of a topological band connects to that of a nearby

band, such that the total Chern number of the bands with connected Hofstadter spectra is zero[41,42]. Consequently, in a Hofstadter energy spectrum for a topological band, the gap with a nonzero Chern number $C$ closes at $\Phi/\Phi_0 \leq 1/|C|$. Indeed, our calculation in Fig. 3a, b confirms this as the CNP gap (0, 0) closes at $\Phi/\Phi_0 < 0.5$, consistent with valley Chern number 2 at the CNP gap of AB-AB TDBG at a finite electric field. This contrasts with the Hofstadter spectrum for AB-BA TDBG at a finite electric field, for which the CNP gap has a zero valley Chern number, and find the CNP gap open throughout $0 < \Phi/\Phi_0 < 1$ (Fig. 4a, b).

To see experimental signatures of nontrivial band topology, we plot $R_{xx}$ as a function of $\nu$ and $B$ for $D/\epsilon_0 = -0.02$ V/nm in Fig. 4c from the 1.10° AB-AB TDBG device. We find that the moiré gap of $\nu = 4$ is

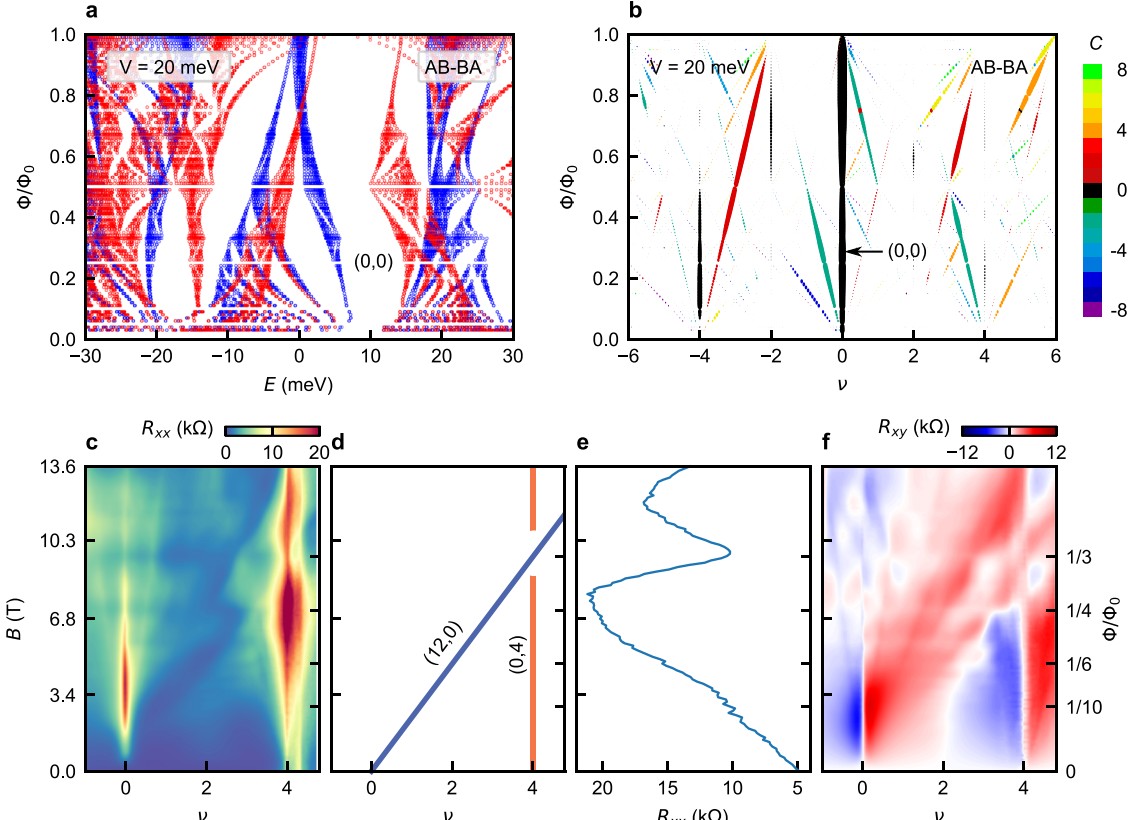

**Fig. 4 | Role of topology on Hofstadter spectra. a** Calculated Hofstadter energy levels for both $K$ (red) and $K'$ (blue) valleys in AB-BA TDBG with twist angle 1.10° for an interlayer potential of $V = 20$ meV. **b** Evolution of gaps corresponding to the spectrum in **a**. Unlike AB-AB case (see Fig. 3a, b), the CNP gap (0, 0) is open for any flux $0 < \Phi/\Phi_0 < 1$. **c** Color-scale plot of $R_{xx}$ as a function $\nu$ and $B$ at $D/\epsilon_0 = -0.02$ V/nm from the 1.10° AB-AB TDBG device. **d** Schematic of **c**, showing the weakening of the moiré gap (0, 4) at $\Phi/\Phi_0 = 1/3$ when a Hofstadter gap (12, 0) crosses. **e** Line slices of $R_{xx}$ vs. $B$ at $\nu = 4$ showing a clear dip at the crossing. **f** Color-scale plot of $R_{xy}$ corresponding to **c**. The Hofstadter gap (12,0) (with finite $R_{xy}$) dominates over the moiré gap (0,4) (zero $R_{xy}$), indicating that Hofstadter spectra across the moiré gap are connected to each other.

weakened at $\Phi/\Phi_0 = 1/3$, as the Hofstadter gap of (12,0) crosses it (also see Fig. 4e for a line-plot of $R_{xx}$ vs. $B$ at $\nu = 4$). The nonzero $R_{xy}$ at the crossing point of the moiré gap, where $R_{xy}$ is otherwise zero, further corroborates the dominance of the Hofstadter gap of (12,0) over the moiré gap of (0,4) (Fig. 4f). This suggests that the Hofstadter spectra across the moiré gap are connected due to the nontrivial topology.

## Discussion

In conclusion, we have presented a comprehensive study of magneto-transport in TDBG in Hofstadter regime, complemented by theoretical calculations of Hofstadter spectra. We identified the manifestation of underlying nontrivial topology of the TDBG flat bands on these spectra. The tunable layer polarization plays a key role in determining the Hofstadter spectra and the quantum phase transition between Chern gaps.

Our central result, that the Chern gap can be controlled by varying the electric field, rather than the charge density, has important implications for magnetoelectric coupling: a Chern gap $\Delta_g$ with Chern number $C$ gives rise to a change in magnetization, $\delta M \propto C\Delta_g$[43]. Thus the physics we have identified allows electrical control of the system magnetization. Here we note that the control over Chern states has been recently demonstrated in twisted monolayer-bilayer graphene by tuning the charge density[44] and in hBN-aligned ABC trilayer graphene using the electric field[19]. Our work demonstrates a novel pathway to control Chern states using the electric field without requiring electron correlation as a prerequisite. Furthermore, the Hofstadter platform of TDBG offers a plethora of Chern transitions over a broad region of electric field. It is interesting to speculate that ferroelectric correlations, as seen in recent

experiments[45], could stabilize Chern bands and the physics we discuss in this study even at zero magnetic field.

## Methods

### Device fabrication

To fabricate TDBG devices, we first exfoliated graphene flakes and cut the selected bilayer graphene flake into two halves by using a scalpel made from an optical fiber[36]. The bilayer graphene flakes were chosen based on optical contrast and later confirmed by Raman spectroscopy. We chose exfoliated hBN flakes of 20 nm to 40 nm in thickness, first based on optical inspection of the color and later measured by AFM. Then we made the hBN-BLG-BLG-hBN stack using the standard poly-carbonate (PC) based dry transfer method[46] and dropped on $SiO_2/Si^{++}$ substrate. The twist angle was introduced by rotating the bottom stage while picking up the second half of the bilayer graphene flake. Afterward, we made the top gate by e-beam lithography and depositing Cr/Au by e-beam evaporation. Subsequently, we defined the geometry of the devices by e-beam lithography followed by etching in $CHF_3+O_2$ plasma. Finally, we made 1D edge contact by etching in $CHF_3+O_2$ plasma and then depositing Cr/Pd/Au.

### Transport measurement

We carried out the low-temperature transport measurements at 300 mK in a He-3 insert inside a liquid He flow cryostat under a perpendicular magnetic field from 0 T to 13.6 T. A current of ~10 nA was sent, and the four-probe voltage was measured using lock-in amplifier using low frequency (~13–17 Hz) after amplifying with a preamplifier. The measurement of the magneto-resistance at the CNP gap presented

in Supplementary Fig. 10 was carried out at 20 mK in a dilution fridge upto $B = 12$ T. The charge density $n$ and the perpendicular electric displacement field $D$ were calculated using the formula, $n = (C_{BG}V_{BG} + C_{TG}V_{TG})/e - n_0$ and $D = (C_{BG}V_{BG} - C_{TG}V_{TG})/2 - D_0$. Here $C_{TG}$ and $C_{BG}$ are the capacitance per unit area of the top and the back gate, respectively, $e$ being the charge of an electron. $n_0$ and $D_0$ are the small offsets in the charge density and the electric displacement field. The capacitance values were calculated at first by noting the dielectric thickness and later estimated more precisely using the magneto-transport features such as the positions of the Brown-Zak oscillations on $B$-axis. To avoid artifacts associated with lead asymmetry we symmetrize the longitudinal resistance as $\bar{R}_{xx}(B) = (R_{xx}(B) + R_{xx}(-B))/2$, and antisymmetrize the transverse resistance as $\bar{R}_{xy}(B) = (R_{xy}(B) - R_{xy}(-B))/2$. The longitudinal conductivity $\sigma_{xx}$ and the transverse conductivity $\sigma_{xy}$ were calculated using the formula $\sigma_{xx} = (w/l)\bar{R}_{xx}/(\bar{R}_{xy}^2 + (w/l)^2\bar{R}_{xx}^2)$ and $\sigma_{xy} = \bar{R}_{xy}/(\bar{R}_{xy}^2 + (w/l)^2\bar{R}_{xx}^2)$, respectively. Here, $w$ is the width and $l$ is the length of the Hall bar geometry. In the text, the symmetrized longitudinal resistance and the antisymmetrized transverse resistance are denoted by $R_{xx}$ and $R_{xy}$ for brevity.

## Twist angle determination

We determine the twist angle $\theta$ based on our low-temperature electron transport measurement, using the relation $n_S = 8\theta^2/\sqrt{3}a^2$. Here $n_S$ is the charge carrier density corresponding to the full filling of the moiré band ($\nu = \pm 4$), $a = 0.246$ is the lattice constant of graphene. To determine $n_S$, we locate $\nu = \pm 4$ by tracing the sequence of Landau levels to the $n$-axis at $B = 0$.

## Data availability

The experimental data used in the figures of the main text are available in Zenodo with the identifier https://doi.org/10.5281/zenodo.5653688[47]. Additional data related to this study are available from the corresponding authors upon reasonable request.

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

## Acknowledgements

We thank Justin C W Song, Allan H MacDonald, Ajit C Balram, and G J Sreejith for helpful discussions. We acknowledge Nanomission grant SR/NM/NS-45/2016 and DST SUPRA SPR/2019/001247 grant along with Department of Atomic Energy of Government of India 12-R&D-TFR-5.10-0100 for support. K.W. and T.T. acknowledge support from the Elemental Strategy Initiative conducted by the MEXT, Japan (Grant Number JPMXP0112101001) and JSPS KAKENHI (Grant Numbers 19H05790 and JP20H00354). D.K.M. would like to acknowledge financial support from Agence Nationale de la Recherche (ANR project "Dirac3D") under Grant No. ANR-17-CE30-0023. D.K.M. and H.A.F. acknowledge support from NSF Grant No. DMR-1914451 and the Research Corporation for Science Advancement through a Cottrell SEED award. H.A.F. further acknowledges the support of NSF Grant No. ECCS-1936406, and of the US-Israel Binational Science Foundation (Grant Nos. 2016130 and 2018726). A.K. acknowledges support from the SERB (Govt. of India) via sanction no. ECR/2018/001443, DAE (Govt. of India) via sanction no. 58/20/15/2019-BRNS, as well as MHRD (Govt. of India) via sanction no. SPARC/2018-2019/P538/SL. D.G. acknowledges the use of HPC facility at IIT Kanpur. D.G. acknowledges the CSIR (Govt. of India) for financial support.

## Author contributions

P.C.A., S.S., C., and L.D.V.S. fabricated the devices. P.C.A. and S.S. did the measurements and analyzed the data. S.L. and A.M. helped in fabrication. D.G., D.K.M, H.A.F., and A.K. did the theoretical calculations. K.W. and T.T. grew the hBN crystals. P.C.A., S.S., A.K., and M.M.D. wrote the manuscript with inputs from everyone. M.M.D. supervised the project.

## Competing interests

The authors declare no competing interests.
