## [Peer Review File · Nature Communications]

Reviewers' comments:

Reviewer #1 (Remarks to the Author):

1) The authors report magneto-transport studies of twisted double bilayer graphene. Similar works have been reported (ref.11-15 in the manuscript). Since the manuscript does not report original experiments, the authors should clarify the progress made here.

2) In the opening bold paragraph, the authors claim that "However, the topological properties of these moire bands such as Chern numbers are little explored". This is not true (see refs.18-26 and many more). In fact, it well known that topology of bands is very import for understanding moire systems. The following recent paper is directly relevant to the specific system studied here,

Phase diagram and orbital Chern insulator in twisted double bilayer graphene

Yi-Xiang Wang, Fuxiang Li, and Zi-Yue Zhang

Phys. Rev. B 103, 115201 – Published 1 March 2021

3) For twisted double bilayer graphene, there is strong evidence for spin-polarized state. The authors should address how these new states modify Hofstadter bands, if they are still relevant.

4) As for the "Electric field drives Chern transition in Hofstadter bands" and "...switch their Chern number on at a time as we vary the electric field". Perhaps the author should mark the Chern number in Fig.4 as function of D .

5) All the experimental data supporting Chern gap is a dip in magnetoresistance. It is common practice in the field to have additional confirmation, such as activation measurement that gives estimate of gap size, especially considering this is a follow-up study.

In conclusion, the experiments appear not original, and I do not find the "new" interpretation of the data convincing or sound. I cannot recommend its publication.

Reviewer #2 (Remarks to the Author):

The manuscript of Adak et al. describes a magnetotransport study in twisted double bilayer graphene with twist angles of $\sim 1^\circ$. The study is focused on the evolution of quantum Hall gaps with different Chern numbers as a function of transverse electric field. The study combines theory and experiment and the theoretical and experimental results are in agreement. The main result is

that at a fixed electric field a subset of gaps with certain Chern numbers are more prominent. The study is carefully carried out, and the manuscript accessible to a broad audience.

There are aspects of the manuscript that could desirably be revised or augmented for publication. The narrative invokes the non-trivial band topology rather often, but the result do not clearly highlight it. How would the experimental observations change if e.g. the bands had the same dispersion, but different topology? A good comparison are twisted double bilayer graphene with twist angles close to 180deg.

Figure 2a and 2c data are plotted in a way that makes it difficult for the reader to get much out of it, except for the gaps already marked.

Reviewer #3 (Remarks to the Author):

The authors report their work on 'Electric field drives Chern transition in Hofstadter bands of twisted double bilayer graphene', and try to understand the effect of band topology on Hofstadter physics. Revealing the influence of band topology on the correlation effects and Hofstadter Physics is interesting and important in current twisted moire system studies. However, after go over the whole manuscript, I worry that the conclusion made by the authors could not be well supported by the present data. So, I would not recommend it to be published in Nature Communications unless the authors could provide further critical experimental supports.

Here are several comments :

1, According to previous transport measurements on either twisted bilayer graphene or mono-bilayer graphene, the nontrivial topological Chern bands have been unveiled near the integer fillings. The spontaneous symmetry broken plays an important role there. The authors studied twisted double bilayer graphene, and also finds the Chern bands emanating from the integer fillings, personally, I do not see any progress or advance from the current work as comparing to those work. The authors must address the difference.

2, In Fig.2a and Fig.2b, the authors try to show the displacement field driven the Chern number variation. In order to confirm the readers, the authors use some lines to guide the eyes. However, from the present data, there are several choices on draw the line. For example, for $D \sim -0.02$, near the (0,0), there are extra fan structure flank the Dirac point with positive and negative slope, maybe give (2,0) or (-2,0). Similar operation could be done on larger D. So I do not confirmed that there is an electric field driven Chern number transition, at least by the current presented data.

3, based on question 2, even in other twisted graphene system with magnetic field and strong correlations, similar behavior could be found. I am not confirmed it its unique to twisted double bilayer graphene, and limited to band topology, unless the band topology here means the strong electron interaction driven nontrivial band topology, then it has been well studied in other systems.

4, In Fig.2, the authors try to use the Fan expanding to other bands to claim the band topology contribution, but the signal or evidence is not that solid, especially under the current data quality. In order to claim this, the authors must improve the data or sample quality.

5, I am not quite sure how well of the current data could be well repeated in other twisted angles. How about the twisted double bilayer near the right angle, like 1.2-1.4degreee? There the physics should be clearer especially the presented understanding would base on an insulating state on the integer fillings. Currently, the authors need a magnetic field to trigger such an insulating gap.

We thank the reviewers for their careful review of our manuscript and the helpful comments. The reviewers have helped us in significantly improving the manuscript, making it clearer and more impactful.

Despite the difficulties of the COVID pandemic we have done new experiments and theory calculations. It has taken some time to respond as both the first author and last author had tested positive and have subsequently recovered.

Below, we address the comments/questions of the reviewers:

(Reviewers' comments/questions are in the blue colored italic font and our responses are in the black colored font)

Reviewer #1:

1) The authors report magneto-transport studies of twisted double bilayer graphene. Similar works have been reported (ref.11-15 in the manuscript). Since the manuscript does not report original experiments, the authors should clarify the progress made here.

Reply: We thank the reviewer for the question that helps us to clarify the progress made by our work. In Table 1, we have summarized the main focus of the mentioned references. The works mentioned by the reviewer focus on exploring the correlated insulating states hosted by the flat bands of twisted double bilayer graphene (TDBG). While some of these works report some magneto-transport, there is no systematic study of the Hofstadter physics or the topology and the corresponding role of the electric field. These works use the magnetic field as a knob to explore the evolution of correlated gaps to identify the nature of underlying ground states, such as in identifying spin-polarized ground states. In particular, the electric field tunability of nontrivial topology remains unexplored not only in TDBG but in any graphene-based systems in general. In our work, we study the Hofstadter physics in TDBG (both experimentally as well as theoretically) and unveil Hofstadter gaps where Chern numbers can be tuned by an electric field. We further identify the role of such electric field tunable layer polarization in the physics we observe.

Recently, topological aspects of the flat band systems are of major interest as evident by the exploration of correlated Chern insulator states in twisted bilayer graphene (TBG). It is important to note the distinction between the two systems: TBG and TDBG, as we summarized in Table 2. Unlike TBG, the electric field can tune the band structure in TDBG. In 2D systems, one can change the charge density to tune the Fermi energy which allows accessing different Chern gaps. In contrast to typical 2D systems, such as TBG, in TDBG we can change the Chern number of the Hofstadter gaps using purely the perpendicular electric field. We would like to emphasize that the observation of *electric field tunable* Chern gaps using Hofstadter bands in our work is the first such report in any 2D system to the best of our knowledge.

Work	Focus of the work
G W Burg et al, Correlated Insulating States in Twisted Double Bilayer Graphene. Phys. Rev. L 123 , 197702 (2019).	 • Reports correlated insulators states at $\frac{1}{2}$ filling at zero magnetic field, at $\frac{1}{3}$ and $\frac{3}{4}$ at finite parallel magnetic fields. • Suggests the role of band separation along with band flatness to observe correlated gaps.
C Shen et al, Correlated states in twisted double bilayer graphene. Nature Physics 16 , 520–525 (2020).	 • Reports control of electric field over the correlated gaps. • The measurement under a parallel magnetic field suggests spin-polarized ordering.
P C Adak et al, Tunable bandwidths and gaps in twisted double bilayer graphene on the verge of correlations. Phys. Rev. B 101 , 125428 (2020).	 • Explores band properties from temperature-dependent resistivity. • Quantifies electric field tunable bandgap magnitudes and bandwidths.
Y Cao et al, Tunable correlated states and spin-polarized phases in twisted bilayer–bilayer graphene. Nature 583 , 215–220 (2020).	 • Reports correlated insulator states that are highly sensitive to the electric field as well as the twist angle. • Provides evidence of spin-polarized ground states. • Reports tunable correlated gaps in higher moiré bands.
X Liu et al, Tunable spin-polarized correlated states in twisted double bilayer graphene. Nature 583, 221–225 (2020).	 • Demonstrates a flat band with supporting correlated insulating states that are tunable by the perpendicular electric field. • Reports a phase transition from a normal metal to a spin-polarized correlated state at half-filling.
This work	 • Reports electric field tunable Chern gaps in the Hofstadter bands. • Study role of the electric field tunable layer polarization and the topology on Hofstadter spectra.

Table 1: Summary of the main focus of earlier works in TDBG. None of these earlier experimental works study magneto-transport systematically (e.g., how Hofstadter spectra change with the electric field).

Twisted graphene systems	Twisted bilayer graphene (TBG)	Twisted double bilayer graphene (TDBG)
Constituent materials	Monolayer graphene + monolayer graphene (1+1)	Bilayer graphene + bilayer graphene (2+2)
Electric field tunability of band structure and topology	No	Yes
Requirement of gates	One gate is sufficient to change charge density	Two gates required to independently control charge density and electric field

Table 2: Summary of key differences between two twisted graphene systems: Twisted bilayer graphene and twisted double bilayer graphene.

Action: To clarify the progress made in our work further, we have modified the introductory paragraphs in the revised manuscript. In particular, we noted, “While the earlier experiments in

TDBG have a major focus on electron correlation physics, the tunability of the topological flat bands in the Hofstadter regime is little explored.” To reflect that we have further understood the mechanism of electric field tunability in terms of layer polarization changes, we have modified the title to be “Vertical electric field drives Chern transitions and layer polarization changes in Hofstadter bands”.

*2) In the opening bold paragraph, the authors claim that "However, the topological properties of these moire bands such as Chern numbers are little explored". This is not true (see refs.18-26 and many more). In fact, it well known that topology of bands is very import for understanding moire systems. The following recent paper is directly relevant to the specific system studied here, Phase diagram and orbital Chern insulator in twisted double bilayer graphene
Yi-Xiang Wang, Fuxiang Li, and Zi-Yue Zhang
Phys. Rev. B 103, 115201 – Published 1 March 2021*

Reply: We thank the reviewer for the question and for pointing out the reference, which we have now cited in the revised version. At the outset, we would like to distinguish our work from the past that we combine detailed experiments with our theoretical calculations to confirm our observations. Furthermore, as Reviewer 2 assesses, “the theoretical and experimental results are in agreement.”

While there are some important theoretical works that explore the Chern physics in TDBG, no experimental study of TDBG has focused on this physics so far. Indeed, the rich physics of tunable topology explored in these theoretical works motivate us to experimentally demonstrate the capability of tuning Chern gaps by the electric field. Our work is also timely, as there has been a rapidly growing interest in exploring correlated Chern insulator states in other twisted graphene systems such as twisted bilayer graphene. Since the Chern insulator states explored in TBG are not tunable by the perpendicular electric field, TDBG provides a platform for engineering such topological states.

Action: To remove any potential confusion we have modified the statement in the opening bold paragraph to highlight that our work explores Chern transition physics experimentally, not only just theoretically. To contrast our work from the experiments exploring correlated Chern insulators in TBG, we have included these sentences in the second paragraph of the revised manuscript: “In the physics of Chern insulator states, as also in the quantum Hall physics due to the formation of Landau levels, gaps with different Chern numbers can be accessed by changing the Fermi energy by varying the charge density. In contrast, a pure electrical control, such as the perpendicular electric field, to manipulate the Chern states has not been demonstrated yet.” We have also included the reference suggested by the reviewer in the revised manuscript.

3) For twisted double bilayer graphene, there is strong evidence for spin-polarized state. The authors should address how these new states modify Hofstadter bands, if they are still relevant.

Reply: We thank the reviewer for the thoughtful question. As we discussed in the main text, we find that Chern gaps change sequentially by 1. To reproduce the experimental results we need to

consider a large spin-splitting in our theoretical calculation ($\Delta_s = 2.5$ meV). The spin-splitting is exchange-enhanced, most likely due to the underlying ferromagnetic order of the spin-polarized states. This supports the statement of the reviewer that there is a spin-exchange interaction in TDBG even though there is no fully formed $\nu = 2$ gap in our device with a twist angle of 1.10 degrees.

To address the query of the reviewer, we have now measured a new device with a twist angle of 1.46 degrees where we observe a fully formed $\nu = 2$ correlated gap that separates the spin-polarized states around $|D| \sim 0.3 - 0.4$ V/nm even at a zero magnetic field (see Fig. R1a). In this device also, we see n vs. D plot with Chern number transition in a sequence of 1 - consistent with our earlier device; this highlights the relevance of underlying spin-polarization in enhancing spin-splitting in the Hofstadter subbands. The value of Δ_s indicates that there is spin-exchange interaction even before the fully formed $\nu = 2$ gap is manifested.

Fig. R1: a, Conductance as a function of moiré filling factor (ν) and displacement field (D) for a device with twist angle of 1.46 degrees. Correlated insulator gaps are observed at $\nu = 2$ around $|D| \sim 0.3 - 0.4$ V/nm. **b**, A fan diagram at an electric field where the correlated gap is observed.

4) As for the "Electric field drives Chern transition in Hofstadter bands" and "...switch their Chern number on at a time as we vary the electric field". Perhaps the author should mark the Chern number in Fig.4 as function of D .

Reply: We thank the reviewer for the comment. The physics of switching the Chern number one at a time as we vary the electric field is explored in detail in Fig. 2a. We have also marked the corresponding Chern numbers by different colored lines as indicated in the legend. We reproduce this in Fig. R2a.

In Fig. 4 of the previous manuscript (reproduced in Fig. R2b), we focused on the variation of conductance at a fixed value of $\nu = 0$. While this plot is not useful to extract the Chern number of the underlying zero energy bands, this allows us to explore the physics of multiple closings and reopenings of gaps governed by tunable layer polarization. We have verified the physics in a new device (see Fig. R2c) as well. However, we realized that this part could potentially distract the

reader from the main physics we explore and hence moved it to the Supplementary Information in the revised version.

Fig. R2: **a**, Switching of Chern numbers in the Hofstadter bands by varying the electric field. The Chern numbers corresponding to the gaps marked by vertical lines are summarized in the legend box. The dashed white line shows the location of ν used in plot **b**. **b-c**, Variation of conductance as a function of B and D at the CNP gap from two devices. The accessible magnetic flux ranges in the two devices are different due to different sizes of moire lattices with different twist angles.

Action: To streamline the story we have moved Fig. 4 and the corresponding discussion of the previous manuscript to the Supplementary Information in the revised version.

5) All the experimental data supporting Chern gap is a dip in magnetoresistance. It is common practice in the field to have additional confirmation, such as activation measurement that gives estimate of gap size, especially considering this is a follow-up study.

Reply: We thank the reviewer for this important question. However, we would like to point out that we had already confirmed the existence of the Chern gap by activation measurement and provided the values of the most prominent gaps in Fig. 2d of the last submitted version of the Supplementary Information (also reproduced in Fig. R3). We find the most prominent gaps to have a value of 0.12-0.42 meV.

Fig. R3: Estimation of LL gaps. (a) Variation of σ_{xx} vs. ν for different values of temperature at $B = 0$ T and $D/\epsilon_0 = 0$ V/nm. (b) Temperature dependence of σ_{xx} minima for four different LL gaps from (a). (c) Extraction of LL gap by fitting the linear region in $\ln(\sigma_{xx})$ vs. $1/T$ plots obtained from (b). Solid lines represent the linear fit. (d) Values of extracted LL gaps in meV at some points in $\nu - D$ parameter space at $B = 9$ T.

In conclusion, the experiments appear not original, and I do not find the "new" interpretation of the data convincing or sound. I cannot recommend its publication.

Reply: We thank the reviewer for the comment. However, we believe that our response to the reviewer's questions will convince the new physics found in our work, namely, the experimental observation of electric field tunable Chern gaps in TDBG; this has not been reported in any other 2D systems. In addition, our theoretical calculations explain our experimental work and provide insights into the role of layer polarization. We have provided additional data from new measurements in the revised Supplementary Information to further support our claims.

Reviewer #2:

The manuscript of Adak et al. describes a magnetotransport study in twisted double bilayer graphene with twist angles of ~ 1 deg. The study is focused on the evolution of quantum Hall gaps with different Chern numbers as a function of transverse electric field. The study combines theory and experiment and the theoretical and experimental results are in agreement. The main result is that at a fixed electric field a subset of gaps with certain Chern numbers are more prominent. The study is carefully carried out, and the manuscript accessible to a broad audience.

Reply: We thank the reviewer for nicely summarizing our work. We appreciate the reviewer's assessment that "the study is carefully carried out", and highlighting that our "theoretical and experimental results are in agreement". We also thank the reviewer for noting that "the manuscript is accessible to a broad audience".

There are aspects of the manuscript that could desirably be revised or augmented for publication. The narrative invokes the non-trivial band topology rather often, but the result do not clearly highlight it. How would the experimental observations change if e.g. the bands had the same dispersion, but different topology? A good comparison are twisted double bilayer graphene with twist angles close to 180deg.

Reply: We thank the reviewer for the suggestion to compare two types of twisted double bilayer graphene systems: AB-AB and AB-BA (twisted with an angle close to 180 degrees). We have measured new devices with the AB-BA configuration. In Fig. R4, we show fan diagrams for two devices with the two different configurations at finite electric fields. At a finite electric field, the CNP gap has a nonzero valley Chern number of 2 in the AB-AB case, whereas for AB-BA the CNP gap has a zero Chern number [Koshino, *Phys. Rev. B* **99**, 235406 (2019)]. Interestingly, we find from Fig. R4 that the CNP gap closes at $\Phi/\Phi_0 \sim 1/5$ in the AB-AB system, in contrast to the CNP gap in the AB-BA device, which remains open throughout the range of magnetic fields we measured. This is consistent with the fact that two bands separated by a nontrivial gap of Chern number C have the Hofstadter spectra connected by a gap closing at $\Phi/\Phi_0 < 1/|C|$, whereas the Hofstadter spectra of two bands separated by a trivial gap can remain separated for any flux. The difference in the evolution of the CNP gap with the magnetic field for two different configurations of AB-AB and AB-BA is also reflected in Fig. R5, where we plot the conductance at $\nu = 0$ as a function of D and B . For AB-AB, the conductance at $D = 0$ remains low at high magnetic fields indicating a gap. This is consistent with zero valley Chern number at the CNP gap for AB-AB TDBG at zero electric field. In contrast, for AB-BA, the trend is the opposite, i.e., the conductance is higher around $D = 0$ throughout the range of the magnetic field we measure, indicating gap closing. This is consistent with the nonzero valley Chern number at the CNP gap for AB-BA configuration at zero electric field.

Fig. R4: Understanding the role of topology by comparing fan diagrams of an AB-AB TDBG device vs. an AB-BA TDBG device at finite electric fields. For the AB-AB case, the CNP gap closes at $\Phi/\Phi_0 \sim 1/5$. In contrast, for AB-BA, the CNP gap remains open throughout the magnetic field range we could explore. Indeed, the accessible range of Φ/Φ_0 is higher in the AB-BA device due to the smaller twist angle.

Fig. R5: Comparing the evolution of conductance at the CNP (i.e., $\nu = 0$) as a function of D and B . For the AB-AB case, the conductance is low at $D = 0$ except at low magnetic fields. For the AB-BA case, the conductance at $D = 0$ remains high throughout the range of magnetic field we measured (indicating gap closing).

We also performed detailed calculations to compare the AB-AB and the AB-BA TDBG with the same twist angle of 1.10 degrees at the interlayer potential of $V = 20$ meV (see Fig. R6). While the band structures are quite similar (see Fig. R6a,b), the Chern numbers are different, as correctly pointed out by the reviewer. In Figs. R6c and R6d, we have plotted the corresponding evolution of Hofstadter gaps as a function of ν and V , and we see a prominent difference between the two cases. To point out the role of topology clearly, we have plotted the evolution of Hofstadter gaps as a function of ν and the magnetic flux Φ/Φ_0 at $V = 20$ meV in Figs. R6e and R6f for AB-AB and AB-BA, respectively. Again, we find that the CNP gap in the Hofstadter spectra is open throughout the range $0 < \Phi/\Phi_0 < 1$ for AB-BA, whereas the gap closes at $\Phi/\Phi_0 < 0.5$ for AB-AB.

This is consistent with the valley Chern numbers $C = 2$ for AB-AB and $C = 0$ for AB-BA at the CNP gap at a finite electric field.

Fig. R6: Role of band topology on the Hofstadter spectra. **a & b**, Zero magnetic field band structures at an interlayer potential of $V = 20$ meV for AB-AB (**a**) and AB-BA (**b**) TDBG with a twist angle of 1.1 degrees. The numbers in the bracket denote Chern numbers of the corresponding bands. **c & d**, Hofstadter gaps as a function of ν and V for AB-AB and AB-BA, respectively. **e & f**, Hofstadter gaps as a function of ν and Φ/Φ_0 for AB-AB and AB-BA, respectively. The arrow indicates a clear difference in the evolution of the CNP gap due to the difference in their topology. The gap closes at $\Phi/\Phi_0 < 0.5$ for AB-AB.

Action: In Fig. 4 of the revised manuscript, we have a more focused discussion on the role of topology in the Hofstadter spectra of TDBG. We have further added the calculation of Hofstadter spectra for AB-BA configuration in the revised Fig. 4.

Figure 2a and 2c data are plotted in a way that makes it difficult for the reader to get much out of it, except for the gaps already marked.

Reply: We thank the reviewer for the comment. Our central result that the Chern gaps in the Hofstadter bands of TDBG are tunable with the electric field is true for all the Chern gaps we observe; we want to emphasize that this is not an effect that can be realized by tuning density or doping. However, to avoid distraction we have focused our analysis on the pair of larger gaps in Figs. 2a and 2c of the main manuscript. For reference, in Fig. R7, we have replotted Fig. 2a with other gaps marked as well. To further support our claim, we have provided additional results from new measurements in the revised Supplementary Information.

Fig. R7: Marking Chern numbers corresponding to dips in conductance. The Chern gaps labeled with blue and orange colors are what we emphasized in the main manuscript; here we marked additional Chern gaps with different colors. The numbers in brackets indicate (C, s) .

Action: We have added the plot with more Chern numbers marked in the revised Supplementary Information. We have provided similar analysis for other twist angles as well from new measurements in the revised Supplementary Information.

Reviewer #3:

The authors report their work on 'Electric field drives Chern transition in Hofstadter bands of twisted double bilayer graphene', and try to understand the effect of band topology on Hofstadter physics. Revealing the influence of band topology on the correlation effects and Hofstadter Physics is interesting and important in current twisted moire system studies. However, after go over the whole manuscript, I worry that the conclusion made by the authors could not be well supported by the present data. So, I would not recommend it to be published in Nature Communications unless the authors could provide further critical experimental supports.

Reply: We thank the reviewer for valuable comments and for pointing out the importance of understanding the influence of band topology in twisted moiré systems. We have now performed more experimental measurements on new devices as well as obtained further numerical data to support our claim. The central result that in TDBG the perpendicular electric field changes the Chern gaps is verified for different twist angles across multiple devices.

Here are several comments :

1, According to previous transport measurements on either twisted bilayer graphene or mono-bilayer graphene, the nontrivial topological Chern bands have been unveiled near the integer fillings. The spontaneous symmetry broken plays an important role there. The authors studied twisted double bilayer graphene, and also finds the Chern bands emanating from the integer fillings, personally, I do not see any progress or advance from the current work as comparing to those work. The authors must address the difference.

Reply: We thank the reviewer for the important question that helps us to clearly differentiate our work from existing literature. As the reviewer has correctly pointed out, some recent works (e.g., Refs 13-18 of the revised manuscript) on twisted bilayer graphene (TBG) or mono-bilayer graphene have reported correlated Chern insulators emanating from integer fillings. These gaps are mostly understood as originating from the two sets of four-fold degenerate (i.e., total eight) low-energy flat bands with non-zero Chern numbers (see Fig. R8a). As the reviewer has correctly pointed out, the main role to separate the eight Chern bands is played by spontaneous symmetry breaking due to electron correlations - the flat band is important for realizing the physics. While the magnetic field helps to unveil the physics, the Hofstadter physics is not essential in understanding the origin of those Chern insulator states.

However, in our work we explore Chern gaps owing to Hofstadter physics at high enough magnetic fields such that $\Phi/\Phi_0 \sim 1$ (see Fig. R8b). While correlations in flat bands can play a secondary role in enhancing the gaps, the main role is played by the interplay of the magnetic length scale and the moiré periodicity to induce fractal-like energy spectra. The signature of the Hofstadter physics is prominent in our data as we also observe strong Brown-Zak oscillations. The Hofstadter physics itself does not require the underlying band to be flat. Indeed, our results of electric field tunable Chern gaps in the Hofstadter band are observed for a wide range of twist

angles in TDBG (see Supplementary Note 4 and Supplementary Figs. 4-7 in the revised Supplementary Information).

While in both cases Chern insulator gaps are observed, the main novelty of our work is that we can use the perpendicular electric field to change Chern numbers in TDBG. We emphasize that this is different from accessing different Landau levels by changing doping in graphene-based systems. Indeed, the pure electric field control over Chern gaps in the Hofstadter bands has not been demonstrated in any other system in the existing literature. Unlike in TDBG, this is not possible in TBG since the band structure in TBG cannot be tuned by a perpendicular electric field and this is why experimental works in TBG thus far employ single gates, while in TDBG devices one uses dual gates to independently control the charge density and the electric field.

Fig. R8: **a**, The physics of correlated Chern insulator states in flat bands of TBG. The two sets of four-fold degenerate low-energy flat bands are symmetry broken due to correlations and correlated Chern insulator states appear at a finite magnetic field. The magnetic field plays a secondary role to stabilize the Chern states. **b**, At high magnetic fields, when $\Phi/\Phi_0 \sim 1$, each copy of the underlying eight bands will give rise to a Hofstadter spectrum. Correlation plays a secondary role to enhance the Hofstadter gaps.

Action: To further clarify the progress made in our work we have added a few sentences in the introductory paragraph of the revised manuscript. In particular, we have noted: “In the physics of Chern insulator states, as also in the quantum Hall physics due to the formation of Landau levels, gaps with different Chern numbers can be accessed by changing the Fermi energy by varying the charge density. In contrast, a pure electrical control, such as the perpendicular electric field, to manipulate the Chern states has not been demonstrated yet.”

2, In Fig.2a and Fig.2b, the authors try to show the displacement field driven the Chern number variation. In order to confirm the readers, the authors use some lines to guide the eyes. However, from the present data, there are several choices on draw the line. For example, for $D \sim -0.02$, near

the (0,0), there are extra fan structure flank the Dirac point with positive and negative slope, maybe give (2,0) or (-2,0). Similar operation could be done on larger D. So I do not confirmed that there is an electric field driven Chern number transition, at least by the current presented data.

Reply: We thank the reviewer for the important question. We agree that the data indeed shows many gaps which naturally occur in the fractal structure of the Hofstadter spectra. In Fig. R9, we have identified different gaps with different colors. Our most interesting observation is that the position of gaps with similar magnitude changes on the ν -axis as D is changed: see the set of gaps marked by the same color. This means that the Chern numbers of the gaps change with the electric field. We then focus on two sets of larger gaps, for sake of brevity, extract their Chern numbers, and see how those numbers evolve with the magnetic field. For reference, in Fig. R9, we have marked the other Chern gaps as well.

Similar observations have been made from the data for different twist angles also, as we explained later (in the reply to comment 5 on page 16).

Fig. R9: Evolution of Chern gaps with the electric field. The Chern gaps labeled with blue and orange colors are what we emphasized in the main manuscript; here we marked additional Chern gaps with different colors. The numbers in brackets indicate (C, s) .

Action: We have added this diagram marking other Chern gaps in the revised Supplementary Information along with similar diagrams for other twist angles from new measurements. Also, in the paragraph describing Fig. 2 in the revised manuscript, we have added a sentence: “Besides the marked pair of dips, there are other dips which are also tuned by the electric field.”

3, based on question 2, even in other twisted graphene system with magnetic field and strong correlations, similar behavior could be found. I am not confirmed it its unique to twisted double bilayer graphene, and limited to band topology, unless the band topology here means the strong electron interaction driven nontrivial band topology, then it has been well studied in other systems.

Reply: We thank the reviewer for the question. While we have pointed out the difference between the physics we study in our work and those in other twisted graphene systems in the reply to

question 1 on page 11, here we would also like to reiterate the uniqueness of twisted double bilayer graphene. The band structure and the topology in TDBG are tunable by the electric field, whereas the electric field does not play any significant role in twisted bilayer graphene.

4, In Fig.2, the authors try to use the Fan expanding to other bands to claim the band topology contribution, but the signal or evidence is not that solid, especially under the current data quality. In order to claim this, the authors must improve the data or sample quality.

Reply: To address the reviewer's comment we have made new devices using the state-of-the-art fabrication method (e.g., using graphite local gate) and reproduced our central result of changing Chern numbers of the Hofstadter gaps by the electric field across multiple angles. However, for the higher angles, the crossing point as evidence of the fan expanding to other bands is not accessible with the magnetic field available to us. To provide further support to our claim, we have done more theoretical calculations.

In Fig. R10a, we show the Hofstadter energy spectrum at the interlayer potential of $V = 20$ meV calculated for 1.1 degrees twisted AB-AB TDBG. The corresponding evolution of the gaps is shown in Fig. R10b. For a finite electric field, the CNP gap ($\nu = 0$) of the zero magnetic field band structure of AB-AB TDBG has a valley Chern number of 2. From Fig. R10a-b, we find that the CNP gap under a perpendicular magnetic field closes at $\Phi/\Phi_0 < 0.5$. This is consistent with the fact that two bands separated by a nontrivial gap of Chern number C have the Hofstadter spectra connected by a gap closing at $\Phi/\Phi_0 < 1/|C|$. In contrast, the Hofstadter spectra of two bands separated by a trivial gap can remain separated for any flux. In Fig. R10c-d, we show the corresponding result for AB-BA configuration with the same twist angle and the interlayer potential from our new calculation. We find that the CNP gap remains open throughout the whole range of $0 < \Phi/\Phi_0 < 1$, consistent with the zero valley Chern number of AB-BA TDBG under a finite electric field.

To highlight the signature of topology in our experimental data, we plot a fan diagram showing R_{xx} as a function of ν and B in Fig. R10e for a TDBG device with a twist angle of 1.1 degrees. We see that the moiré gap (0,4) is dominated by the Hofstadter gap (12,0) at $\Phi/\Phi_0 = 1/3$. To clearly visualize this effect, we have plotted a line slice of R_{xx} at the moiré gap as a function of B and see a dip at the crossing point between two gaps. The dominance of the Hofstadter gap over the moiré gap is also visualized from the color-scale plot of R_{xy} , where we see a finite R_{xy} at the crossing point, whereas R_{xy} is otherwise zero at the moiré gap. These results establish that the Hofstadter spectra of the two bands separated by the moiré gap are connected - a manifestation of the nontrivial topology.

Fig. R10: **a,c**, Hofstadter spectra at the interlayer potential of $V = 20$ meV for AB-AB and AB-BA configurations, respectively. **b,d**, Evolution of Hofstadter gaps corresponding to **a** and **c**, respectively. **e**, Fan diagram at $D/\epsilon_0 = -0.02$ V/nm for AB-AB device with twist angle 1.1 degrees. **f**, Schematic depicting the moiré gap (0,4) intersected by the Hofstadter gap (12,0). **g**, Line slice of resistance from **e** at $\nu = 4$ as a function of the magnetic field showing a dip in the moiré gap resistance at $\Phi/\Phi_0 = 1/3$. **h**, Color-scale plot of R_{xy} corresponding to the fan diagram in **e** showing that R_{xy} is finite at the intersection of the moiré gap (0,4) with the Hofstadter gap (12,0). R_{xy} at the moiré gap is otherwise zero.

Action: To further corroborate our experimental observation of the signature of nontrivial band topology on the Hofstadter spectra, we have provided new calculations in Fig. 4 of the revised manuscript. Also, to visualize the signature of topology in our experimental data more clearly, we have provided the color-scale plot of R_{xx} along with the line slice in the revised Fig. 4.

5, I am not quite sure how well of the current data could be well repeated in other twisted angles. How about the twisted double bilayer near the right angle, like 1.2-1.4degreee? There the physics should be clearer especially the presented understanding would base on an insulating state on the integer fillings. Currently, the authors need a magnetic field to trigger such an insulating gap.

Reply: We thank the reviewer for asking this question. As per the suggestion of the reviewer, we verified the repeatability of our results for different twist angles by making new devices. In Fig. R11a, we show the evolution of conductance as a function of the filling factor and the electric field for a twist angle of 1.46 degrees. Indeed, for this twist angle, we see a clear correlated gap at $\nu = 2$ at the zero magnetic field (see Fig. R11b for a line slice). Fig. R11c shows the evolution of conductance at 12.5 T. Again, we observe multiple Hofstadter gaps that change their Chern numbers with the perpendicular electric field. Similar observations have been made for other twist angles as well as included in the revised Supplementary Information.

Fig. R11: **a**, Conductance as a function of filling factor and the electric displacement field at zero magnetic field for twist angle 1.46 degrees. A correlated gap appears at $\nu = 2$ for $|D| \sim 0.4$ V/nm. **b**, A line slice at $D = -0.49$ V/nm clearly showing a peak in the resistance at $\nu = 2$. **c**, Evolution of conductance at a finite magnetic field of 12.5 T showing multiple peaks/dips corresponding to the Chern gaps evolving with the electric field. **d**, (C, s) values corresponding to the σ_{xx} dips in **c**.

Action: We have added the results from new measurements for different twist angles in the revised Supplementary Information. This establishes the repeatability of our observation even for the angles that the reviewer points out where an explicit $\nu = 2$ gap is seen at the zero magnetic field.

Reviewers' comments:

Reviewer #1 (Remarks to the Author):

The authors have made significant revisions to the manuscript and added new experimental data. Their responses to reviewers' comments are reasonable. Now I recommend its publication in Nature Communications.

Reviewer #2 (Remarks to the Author):

The revised manuscript includes some improvements from the previous version. However, the experimental data remains perhaps not entirely compelling. The main finding is the observation of a set of quantum Hall states (QHSs) which originate at half moire Brillouin zone (BZ) filling ($s=-2$), which evolve with the transverse electric field. The authors explain theoretically the findings using a model where the both the valley and spin degrees of freedom are no longer degenerate. The valleys are split by the electric field, and the spin through the use of a Zeeman splitting in the model. The calculations identify gaps at fixed energy that change Chern number as the electric field is varied.

However the identification of the $s=-2$ QHS is not too convincing. The lines marking the various $s=-2$ QHSs in Fig. 2b are fitted over a narrow range of magnetic field, and the slope may have uncertainty. In some cases, it's not clear that a line can be drawn with any accuracy through the data, as it's the case for (1,-2) state in Fig. 2b. Moreover, a distinctive element in the data is that most $s=-2$ QHS have positive Chern numbers, while previous studies in TDBG (arXiv:2109.08255) or TBG observe QHSs with primarily, though not exclusively, negative Chern numbers for $s<0$, and positive Chern numbers for $s>0$. Lastly, based on what is known on the electrostatics of TDBG (DOI:10.1126/science.abc3534) the range of on-site layer energies used in the simulations appears high compared to the electric field used experimentally.

Some of the concluding statements are not entirely consistent with the findings. For examples the authors state "In the physics of Chern insulator states, as also in the quantum Hall physics due to the formation of Landau levels, gaps with different Chern numbers can be accessed by changing the Fermi energy by varying the charge density. In contrast, a pure electrical control, such as the perpendicular electric field, to manipulate the Chern states has not been demonstrated yet." The data shows that one have to change both the electric field and the density in order to change the Chern number. In the calculations the Chern gaps evolve with electric field at a fixed energy. In the

experimental data at a fixed magnetic field (flux/moire unit cell) QHS with different Chern numbers are observed at different electric field AND different moire Brillouin zone filling (ν).

Overall the findings are interesting and probably correct, although the experimental data is not entirely convincing.

Reviewer #3 (Remarks to the Author):

I would appreciate the authors' effort on addressing my comments. However, I do not think my concerns/worries are fixed. As the authors could read from my last turn comments, the qualities of the experimental data could not well support their argument. So at that time, I do not recommend it to be published in Nature Communications. Now, even though the authors try to re-analyze their data again for better supporting, but the critical drawback is still there. I would conclude that this is only the preliminary data, not enough on getting the conclusion. Without the solid new data support, I will not recommend it to be published in Nature Communications.

We thank the reviewers for reviewing our manuscript. Below, we respond to the reviewers' comments/questions on the resubmitted manuscript:

(Reviewers' comments/questions are in the *blue colored italic font* and our responses are in the black colored font.)

Reviewer #1:

The authors have made significant revisions to the manuscript and added new experimental data. Their responses to reviewers' comments are reasonable. Now I recommend its publication in Nature Communications.

We thank the reviewer for reviewing our manuscript and recommending the publication of the revised manuscript in Nature Communications. We are happy to note that the reviewer finds that we have reasonably addressed all the reviewers' comments.

Reviewer #2:

The revised manuscript includes some improvements from the previous version. However, the experimental data remains perhaps not entirely compelling. The main finding is the observation of a set of quantum Hall states (QHSs) which originate at half moire Brillouin zone (BZ) filling ($s=-2$), which evolve with the transverse electric field. The authors explain theoretically the findings using a model where the both the valley and spin degrees of freedom are no longer degenerate. The valleys are split by the electric field, and the spin through the use of a Zeeman splitting in the model. The calculations identify gaps at fixed energy that change Chern number as the electric field is varied.

However the identification of the $s=-2$ QHS is not too convincing. The lines marking the various $s=-2$ QHSs in Fig. 2b are fitted over a narrow range of magnetic field, and the slope may have uncertainty. In some cases, it's not clear that a line can be drawn with any accuracy through the data, as it's the case for (1,-2) state in Fig. 2b. Moreover, a distinctive element in the data is that most $s=-2$ QHS have positive Chern numbers, while previous studies in TDBG (arXiv:2109.08255) or TBG observe QHSs with primarily, though not exclusively, negative Chern numbers for $s<0$, and positive Chern numbers for $s>0$.

Reply: We thank the reviewer for comparing our results with previous studies. However, we would like to point out that a fractal energy spectrum in Hofstadter physics contains numerous gaps with both positive and negative Chern numbers. The gaps disperse on the fan diagram along linear trajectories governed by the Diophantine equation for all possible combinations of integers (C, s) (see the gray lines in Fig. R1b),

$$\nu = C \frac{\phi}{\phi_0} + s. \quad (\text{Eq. R1})$$

Depending on the actual energy spectrum, some gaps are larger than others and experimentally observed more clearly (e.g., colored lines in Fig. R1b). On the other hand, the QHSs observed in TBG are *correlated* Chern insulator gaps, which occur due to correlation-driven symmetry breaking of four-fold degenerate conduction and valence flat bands (total eight). As depicted in Fig. R1c, conduction and valence Chern bands have Chern numbers of magnitude 1 but with opposite signs and thus explain negative C for $s < 0$ and positive C for $s > 0$ (Fig. R1d). Figs. R1b and R1d contrast the two physics.

Fig. R1. Comparison between Hofstadter physics and correlated Chern insulator physics. a, Example of a Hofstadter energy spectrum. **b,** The Hofstadter gaps evolve along linear trajectories (gray lines) governed by Diophantine Eq. R1. In experimental data, larger gaps are resolved (colored lines). [See B. Hunt et al, *Science* **340**, 1427 (2013) for example]. **c,** Origin of correlated Chern insulator in TBG. **d,** Evolution of correlated Chern insulator gaps in fan diagram [see I Das et al, *Nature Physics* **17**, 710 (2021), for example].

Fig. R1b also explains how the dispersions of the Hofstadter gaps in the fan diagram are bound by the rational value of Φ/Φ_0 . This is because at magnetic flux commensurate with the superlattice periodicity, the effective magnetic field felt by electrons is zero resulting in conductivity peaks at $\Phi/\Phi_0 = 1/2, 1/3, 1/4, \dots$, known as Brown-Zak oscillations. The observation of Brown-Zak oscillations in our data not only corroborates the Hofstadter physics but also explains why some Hofstadter gaps being smaller in size are observed over a narrower range of magnetic fields.

Fig. R2. Details of fitting to extract (C, s) . **a**, Fan diagrams reproduced from Fig 2b of the main manuscript. **b**, σ_{xx} vs ν plots where magnetic field (B) is varied in steps of 0.1 T, showing clear minima in σ_{xx} corresponding to different (C, s) states as indicated by the arrows. The three different plots are for the three different electric fields used in **a**. Within a particular subpanel, each σ_{xx} vs ν plot is shifted up by 0.5 units. **c**, Fitting of the extracted (ν, B) points of σ_{xx} minima for the different (C, s) states. The blue (orange) dots indicate σ_{xx} minima for $s = -2$ ($s = 0$) state. The extracted (C, s) values are indicated in each plot. The corresponding σ_{xx} vs ν plots from which the σ_{xx} minima for $s = -2$ states are extracted are shown in **b**. Thirty-one σ_{xx} vs ν slices are used to fit each $(C, s = -2)$ state

To further address the reviewer's comment about the fitting and its uncertainty, we show the analysis of extracting (C, s) of Hofstadter gaps in Fig. R2 in detail. In Fig. R2b, we plot multiple

line slices of σ_{xx} vs. ν at different values of B and identify the location of minima corresponding to the gaps. We then plot the (ν, B) points of the minima in Fig. R2c and fit them with straight lines. As shown in Fig. R2c, the extracted C values indicate low uncertainty in C (for example, the extracted value of the $(1, -2)$ state pointed out by the reviewer comes out to be $(0.99, -2.00)$). We further note that even for $s = -2$ Chern gaps our fitting is robust across a magnetic field range of 3 T. For references, we include some data in Fig. R3 from other papers on TDBG (mentioned by the reviewer) and TBG, which also report Chern gaps over similar ranges of magnetic field (for example see the $(3,1)$ state in Fig. R3c).

Fig. R3. Data from literature as reference. **a**, Observation of correlated Chern insulators in TBG as reported in *I Das et al, Nature Physics 17, 710 (2021)*. **b**, Correlated Chern insulators in TBG as reported in *S Wu et al, Nature Materials 20, 488 (2021)*. **c**, Chern gaps as observed in *M He et al, arXiv:2109:08255*.

Action: We have included the fitting details in the revised Supplementary Information.

Lastly, based on what is known on the electrostatics of TDBG (DOI:10.1126/science.abc3534) the range of on-site layer energies used in the simulations appears high compared to the electric field used experimentally.

Reply: We appreciate the reviewer’s comment. First of all, we would like to emphasize that the values of different hopping parameters used in the band structure calculation of TDBG are not fixed yet and vary across the literature (we use the values from *Koshino et al. Phys. Rev. B 99, 235406 (2019)*). Therefore, while the theoretical calculation in TDBG helps to understand the underlying physics, an exact quantitative match between experiment and theory is not expected. In our work, the central result that the Hofstadter gaps change their Chern numbers with the electric field is valid for any value of the electric field. When we use slightly higher values of

interlayer potential in our calculation, it turns out that even the sequence of the changes in the calculated Chern number matches well with the experiment.

We further add, a relation between the on-site layer potential V and the electric displacement field D can be written as

$$V = eDd/\epsilon_0\epsilon = bD/\epsilon_0. \quad (\text{Eq. R2})$$

Here, e is the electron charge, d is the distance between graphene layers, and ϵ is the dielectric constant. By putting $d = 0.33$ nm and $\epsilon = 4$, a typical value of 82.5 meV/(V/nm) is obtained for the factor $b = ed/\epsilon$. The mentioned reference also states a similar value of the factor $b \approx 59$ meV/(V/nm). The conversion, however, depends on the detail variation of ϵ which changes with the electric field. For example, the mentioned reference itself shows how the dielectric function can change its value by one order of magnitude as the interlayer potential is increased from 0 to 40 meV (Supplementary Figure S20 in DOI:10.1126/science.abc3534). Additionally, the extent of the screening effect also varies with the twist angle. The variation of screening with the electric field and twist angle could further explain high onsite layer potentials corresponding to the experimental electric field.

Some of the concluding statements are not entirely consistent with the findings. For examples the authors state "In the physics of Chern insulator states, as also in the quantum Hall physics due to the formation of Landau levels, gaps with different Chern numbers can be accessed by changing the Fermi energy by varying the charge density. In contrast, a pure electrical control, such as the perpendicular electric field, to manipulate the Chern states has not been demonstrated yet." The data shows that one have to change both the electric field and the density in order to change the Chern number. In the calculations the Chern gaps evolve with electric field at a fixed energy. In the experimental data at a fixed magnetic field (flux/moire unit cell) QHS with different Chern numbers are observed at different electric field AND different moire Brillouin zone filling (ν).

Reply: We thank the reviewer for this comment. The trajectory of a Chern gap in the parameter space of (ν, B) is governed by the Streda formula,

$$\frac{\partial n}{\partial B} = C \frac{e}{h}. \quad (\text{Eq. R3})$$

Thus, when C is varied by changing any parameter (in our experiment D), for a fixed value of B , the location of the Chern gap on n -axis (equivalently, on ν -axis) also changes. Therefore, though showing the change in C by the electric field at fixed energy is straightforward in the calculation, it involves a change in n as well in the experiment. This is depicted schematically in Fig. R4a-b. The role of the electric field is reflected in the finite range of D for which different Chern gaps are accessible. Specifically, using a pure electric field at fixed density we can open up a Chern insulating state from a bulk gapless state. In contrast, in simple Landau level physics, different Chern gaps are present at different energies and are unchanged for any value of the electric field (see R4c).

Action: We have modified line number 23 of the main manuscript to give the message more clearly.

Fig. R4. a, Schematic representation of the origin of electric field tunable Chern gaps. Chern number of the gaps changes when Hofstadter subbands peel off from a group of Hofstadter subbands and merge with another group as the electric field is varied. **b**, A depiction of gap evolution on charge density-electric field-diagram corresponding to **a**. **c**, In simple Landau level physics different Landau levels are at constant charge density without being modified by the electric field.

Overall the findings are interesting and probably correct, although the experimental data is not entirely convincing.

Reply: We thank the reviewer for finding our result interesting. We also draw the review's attention to the fact that our experimental results are consistent across multiple devices. We hope that our response establishes that the experimental data convincingly support our findings.

Reviewer #3:

I would appreciate the authors' effort on addressing my comments. However, I do not think my concerns/worries are fixed. As the authors could read from my last turn comments, the qualities of the experimental data could not well support their argument. So at that time, I do not recommend it to be published in Nature Communications. Now, even though the authors try to re-analyze their data again for better supporting, but the critical drawback is still there. I would conclude that this is only the preliminary data, not enough on getting the conclusion. Without the solid new data support, I will not recommend it to be published in Nature Communications.

Reply: We thank the reviewer for the comment. However, it is not clear to us what data quality the reviewer refers to. The electric field tunability of the Chern gap evolution, which is the central result of our work, is self-evident from Fig. 2a in the main manuscript – the location of Chern gaps (i.e, σ_{xx} dips) clearly changes with the electric field. We further extract the Chern numbers by careful analysis – though all the dips might not be visible in a single *color-scale plot*, we can clearly track the σ_{xx} -minima from the line slices and extract the (C, s) values with low uncertainty as explained in Fig. R2. Furthermore, we would like to draw the reviewer's attention that in the last review response, we did not merely re-analyze old data, but provided new data from the measurement of new devices. That the electric field changes Chern numbers of the Hofstadter gaps is observed across multiple devices and also independently verified by our theoretical calculation.

REVIEWER COMMENTS

Reviewer #2 (Remarks to the Author):

I appreciate the authors response, although they may have misunderstood the main question related to the quantum Hall states Chern numbers.

Each broken-symmetry correlated insulator (CI) in either twisted bilayer graphene (TBG) or twisted double bilayer graphene (TDBG) should generate a fan of quantum Hall states with both positive and negative Chern numbers, irrespective of the Chern number of the correlated insulator. However, experimentally the Landau fans tend to point away from charge neutrality in both TBG and TDBG, equivalent to Landau fans have predominantly positive (negative) Chern numbers for positive (negative) moiré unit cell filling. In TBG the CIs also have a non-zero Chern number, but in TDBG the CIs at ± 2 electrons per moiré unit cell appear to have a zero Chern number (arXiv:2109.08255). So the experimental observation I was referring to, where Landau fan originating from CIs tend to point away from neutrality is not necessarily dependent on whether the CI is topologically non-trivial or not. In TBG these experimental observations have been theoretically explained by an interaction effect that leads to a smaller (larger) effective mass for excitations away (towards) charge neutrality (Phys. Rev. Lett. 127, 266402, 2021). But experimentally Landau fans in both TBG and TDBG appear to have the same trend, pointing away from neutrality.

That said my statement is based on prior experimental observations, but in principle a quantum Hall state with positive Chern number at negative moiré unit cell filling as reported in the manuscript is not precluded, and I trust the authors have exercised due diligence in data analysis.

Reviewer #3 (Remarks to the Author):

In the first turn, I think I have listed all the details in my review comments. My most concerns are the experimental data cannot support the authors' main claims. In the reply, I did not see any experimental new data added in to improve the paper/data quality. Actually, when I go over another referee's comments on this work, same issue listed there. In the current status, the experimental data is quite preliminary, I am not convinced by their data. Once again, I do not think it is worth to be considered for publication in Nature Communications unless enough and solid experimental data are provided.

Reviewer #4 (Remarks to the Author):

The manuscript by Pratap Chandra Adak et al. reports the study of Chern insulating states in twisted double-bilayer graphene. The authors focus on the Chern insulating states that emerge at high magnetic fields in Hofstadter bands. They use the measurements of the longitudinal resistance as a function of the band filling, displacement field, and magnetic field to make statements regarding these Chern states. While I think some of the data in the paper might be interesting, I can not recommend the paper for publication in Nature Communication for the reasons stated below.

1) The authors emphasize in their abstract that “However, the topological properties of these moiré bands such as Chern numbers are little explored experimentally.” I could not agree with this: investigating the topology properties and measuring Chern numbers of the flat moiré bands has been one of the major directions and has already resulted in many exciting discoveries. For example, the discoveries of zero-field Chern insulators (QAHE) in the twisted bilayer and ABC trilayer aligned to hBN, twisted monolayer-bilayer graphene, and recently in twisted TMDs. The Hofstadter Chern insulating states in twisted bilayer were also studied in detail by multiple groups (Ref. 13 and many others).

2) The authors also put an emphasis on the ability to electrically tune Chern states. For example, in line 23: “In contrast, a pure electrical control, such as the perpendicular electric field, to open up a Chern insulating state from a bulk gapless state has not been demonstrated yet.”

This statement is incorrect. For example, D-field tunable crossings between Landau levels in twisted bilayer graphene were observed ten years ago (PRL 108, 076601 (2012), Sanchez-Yamagishi et al.)

Moreover, electric-field-tunable Chern insulators were observed even at zero field, for example, in ABC trilayer on hBN (Chen et al. Nature 579 (2020)) and in twisted monolayer-bilayer graphene (Polshyn et al. Nature 588 (2020)). In the case of ABC on hBN, it was shown that the electrical field alone could be used to switch between $C=2$ and $C=0$ states. The authors did not even cite this most relevant paper.

To conclude, I don't see very significant progress or novelty in the results that the author present in their manuscript. The electrical switching of Chern states has already been demonstrated in multiple other systems, and there it happens in more interesting contexts.

3) Finally, I also somewhat agree with the other reviewer regarding the quality of the data. For example, in Fig.2a, half of the marked Chern insulating states are blurred to the extent that they are

merging together. Furthermore, the authors say that they could not observe quantization of the Hall conductance for the Chern states explaining it by the disorder. At this point, providing both longitudinal and transverse resistance became the standard in the field. Other groups routinely observe quantized or approximately-quantized Hall resistance for Chern insulating states in both high and zero magnetic fields.

To sum it up, I think that the paper lacks sufficient novelty or a major advance. It also makes several questionable statements. Hence, I cannot recommend the manuscript for publication in Nature Communication.

We thank the reviewers for reviewing our manuscript. Below, we respond to the reviewers' comments/questions:

(Reviewers' comments/questions are in *blue colored font* and our responses are in black colored font.)

Reviewer #2 (Remarks to the Author):

I appreciate the authors response, although they may have misunderstood the main question related to the quantum Hall states Chern numbers.

Each broken-symmetry correlated insulator (CI) in either twisted bilayer graphene (TBG) or twisted double bilayer graphene (TDBG) should generate a fan of quantum Hall states with both positive and negative Chern numbers, irrespective of the Chern number of the correlated insulator. However, experimentally the Landau fans tend to point away from charge neutrality in both TBG and TDBG, equivalent to Landau fans have predominantly positive (negative) Chern numbers for positive (negative) moiré unit cell filling. In TBG the CIs also have a non-zero Chern number, but in TDBG the CIs at ± 2 electrons per moiré unit cell appear to have a zero Chern number (arXiv:2109.08255). So the experimental observation I was referring to, where Landau fan originating from CIs tend to point away from neutrality is not necessarily dependent on whether the CI is topologically non-trivial or not. In TBG these experimental observations have been theoretically explained by an interaction effect that leads to a smaller (larger) effective mass for excitations away (towards) charge neutrality (Phys. Rev. Lett. 127, 266402, 2021). But experimentally Landau fans in both TBG and TDBG appear to have the same trend, pointing away from neutrality.

That said my statement is based on prior experimental observations, but in principle a quantum Hall state with positive Chern number at negative moiré unit cell filling as reported in the manuscript is not precluded, and I trust the authors have exercised due diligence in data analysis.

Reply: We thank the reviewer for the careful review and for nicely summarizing the trend of Landau levels/Chern insulators in TBG and TDBG. Indeed, unlike usual Landau levels/Chern insulators, the Chern states in the Hofstadter regime evolve in both directions - toward or away from the charge neutrality; we see this in our experiment. This is primarily because energy levels with both positive and negative Chern numbers are intertwined in the fractal of Hofstadter spectra. We see this in our theoretical calculations as well. We also appreciate the reviewer for noting our careful data analysis.

Complete understanding of Hofstadter physics in flat bands is ongoing work; we humbly believe that our observations in TDBG will initiate further exploration of electric-field tuning Chern numbers using Hofstadter physics.

Reviewer #3 (Remarks to the Author):

In the first turn, I think I have listed all the details in my review comments. My most concerns are the experimental data cannot support the authors' main claims. In the reply, I did not see any experimental new data added in to improve the paper/data quality. Actually, when I go over another referee's comments on this work, same issue listed there. In the current status, the experimental data is quite preliminary, I am not convinced by their data. Once again, I do not think it is worth to be considered for publication in Nature Communications unless enough and solid experimental data are provided.

Reply: We appreciate the reviewer for his/her valuable time reviewing our manuscript. We would like to point out that not only our experimental results are consistent across multiple devices with twist angles 1.09° , 1.10° , 1.42° , and 1.46° (e.g., see Supplementary Figures 7-9), our independent theoretical calculations firmly support our observations. We have added a new figure in the revised Supplementary Information that shows σ_{xy} values mostly consistent with the corresponding Chern numbers. Hence, our observation of σ_{xx} minima along with the σ_{xy} data provides strong evidence of the (C, s) Chern states. Though the quantization in Hall conductance is somewhat weak, we humbly note that the earlier studies of TDBG (e.g., refs 28-32) have not shown a clear quantization in their magneto-transport data, despite the same groups exploring well-quantized correlated Chern insulator states in TBG. On page 4, while addressing the comments of Reviewer 4, we have further discussed the possible reasons for weak quantization.

Action: In the revised Supplementary Information, we have now included the details of the Hall conductance data (Supplementary Fig. 5).

Reviewer #4 (Remarks to the Author):

The manuscript by Pratap Chandra Adak et al. reports the study of Chern insulating states in twisted double-bilayer graphene. The authors focus on the Chern insulating states that emerge at high magnetic fields in Hofstadter bands. They use the measurements of the longitudinal resistance as a function of the band filling, displacement field, and magnetic field to make statements regarding these Chern states. While I think some of the data in the paper might be interesting, I can not recommend the paper for publication in Nature Communication for the reasons stated below.

Reply: We thank the reviewer for summarizing our manuscript and appreciating some of the data. Below we address his/her comments.

1) The authors emphasize in their abstract that “However, the topological properties of these moiré bands such as Chern numbers are little explored experimentally.” I could not agree with this: investigating the topology properties and measuring Chern numbers of the flat moiré bands has been one of the major directions and has already resulted in many exciting discoveries. For

example, the discoveries of zero-field Chern insulators (QAHE) in the twisted bilayer and ABC trilayer aligned to hBN, twisted monolayer-bilayer graphene, and recently in twisted TMDs. The Hofstadter Chern insulating states in twisted bilayer were also studied in detail by multiple groups (Ref. 13 and many others).

Reply: We thank the reviewer for the comment. We agree with the reviewer that many exciting discoveries including (correlated) Chern Insulator states have resulted from studying different twisted systems (e.g., refs 13-18 in our manuscript). However, we would like to point out that all these works report the observations of Chern insulator states as a consequence of electron correlation. In those studies, the correlation effect facilitates symmetry breaking to lift the degeneracy of four-fold degenerate flat bands. Indeed, without the correlation effect, ‘zero-field’ Chern insulators cannot arise since Chern insulator states require the breaking of time-reversal symmetry. Otherwise, opposite Chern numbers from two opposite valleys cancel each other.

While observations of correlated Chern insulators are novel due to the possibility of zero-field Chern insulators, an important question remains – how the underlying topology (as indicated by nonzero valley Chern number) of the flat bands manifests without any role of correlations. However, implications of the flat bands’ topology in a magnetic field within the framework of single-particle physics have not received much attention. This is what we meant in the two sentences: “To date, the major focus has been to gain new insights into correlated-electron physics hosted by the flat bands. However, the topological properties of these moiré bands such as Chern numbers are little explored experimentally.” Our work explores that direction as we study the implication of underlying band topology on its Hofstadter physics, which does not exclusively require correlations. For example, we demonstrate the perpendicular electric field tunable Chern gaps in devices without (Fig. 2a of the main manuscript, and Supplementary Fig. 7b) and with (Supplementary Fig. 8d and 9b) a correlated insulator feature at $\nu=2$. The prospect of interplay with correlations enriches physics further.

Action: In view of the reviewer’s comment, to deliver the intended message more appropriately we modified the sentence in the abstract of the revised manuscript: “However, the implications of the topology of these moiré bands within the framework of single-particle physics are little explored experimentally.”

2) The authors also put an emphasis on the ability to electrically tune Chern states. For example, in line 23: “In contrast, a pure electrical control, such as the perpendicular electric field, to open up a Chern insulating state from a bulk gapless state has not been demonstrated yet.”

This statement is incorrect. For example, D-field tunable crossings between Landau levels in twisted bilayer graphene were observed ten years ago (PRL 108, 076601 (2012), Sanchez-Yamagishi et al.)

Moreover, electric-field-tunable Chern insulators were observed even at zero field, for example, in ABC trilayer on hBN (Chen et al. Nature 579 (2020)) and in twisted monolayer-bilayer graphene (Polshyn et al. Nature 588 (2020)). In the case of ABC on hBN, it was shown that the electrical field alone could be used to switch between $C=2$ and $C=0$ states. The authors did not even cite this most relevant paper.

To conclude, I don't see very significant progress or novelty in the results that the author present in their manuscript. The electrical switching of Chern states has already been demonstrated in multiple other systems, and there it happens in more interesting contexts.

Reply: We thank the reviewer for the comment. We are aware of the study of D-field tunable Landau level crossings in twisted bilayer graphene by Sanchez-Yamagishi et al. *PRL* **108**, 076601 (2012), which we had cited in the Supplementary Information. Previous works from our group also explored the physics of Landau level crossing in ABA-trilayer graphene in detail (*Nat. Comm.* **8**, 14518 (2017), *PRL* **121**, 056801 (2018)). However, Landau level crossings do not necessarily change the Chern number (these crossings occur between Landau levels with the same Chern number). For example, Sanchez-Yamagishi et al. studied the change in layer polarization to spin-polarization state by Landau level crossing due to D-field; there was no change in Chern number as a function of D-field. Similar work was done in bilayer graphene (BLG) as well (e.g., *Science* **330**, 812 (2012)).

While the work by Polshyn et al. *Nature* **588**, 66 (2020) explored the electrical control of Chern states, the control was achieved by changing the charge density, not the pure electric field. We appreciate the reviewer for highlighting the relevance of the important work by Chen et al. *Nature* **579**, 56 (2020). While the electric field tunability of correlated Chern insulator states studied by Chen et al. is novel, we believe that our demonstration is also significant for not requiring correlations as a prerequisite. Furthermore, the electric field tunability of Chern numbers is abundant in our system since Hofstadter spectra naturally possess plenty of subbands with many different Chern numbers.

Action: In the revised manuscript, we have included the mentioned references in appropriate locations. To further clarify the novel findings in our work, we have added the following sentences in the discussion of the revised manuscript: "Here we note that the control over Chern states has been recently demonstrated in twisted monolayer-bilayer graphene by tuning the charge density and in hBN-aligned ABC trilayer graphene using the electric field. Our work demonstrates a novel pathway to control Chern states using the electric field without requiring electron correlation. Furthermore, the Hofstadter platform of TDBG offers a plethora of Chern transitions over a broad region of electric field."

3) Finally, I also somewhat agree with the other reviewer regarding the quality of the data. For example, in Fig.2a, half of the marked Chern insulating states are blurred to the extent that they are merging together. Furthermore, the authors say that they could not observe quantization of the Hall conductance for the Chern states explaining it by the disorder. At this point, providing both longitudinal and transverse resistance became the standard in the field. Other groups routinely observe quantized or approximately-quantized Hall resistance for Chern insulating states in both high and zero magnetic fields.

Reply: We thank the reviewer for the comment. In Fig. 2a, we plot the longitudinal conductance in a color-scale plot to show the plethora of tunable Chern states. Though it helps to visualize multiple Chern states all at once, we agree that some individual states look blurred. However, as

we showed in Supplementary Fig. 6, the minima corresponding to Chern gaps are prominent in the line slices and we can trace them over a large range of magnetic field to extract the corresponding Chern numbers unambiguously.

In Fig. R1, we plot the Hall conductance as a function of the filling across Chern gaps. We do observe approximately-quantized Hall conductance. For example, the measured values of σ_{xy} at the location of σ_{xx} minima corresponding to different Chern gaps (shaded with blue-colored backgrounds) are very close to the quantized value of Ce^2/h (indicated by dashed lines).

Fig. R1: Hall conductance of (C, s) states. **a**, Fan diagrams reproduced from Fig. 2b of the main manuscript, with solid lines overlaid on σ_{xx} minima, indicating the (C, s) Chern gaps. **b**, **c**, Line slices of σ_{xx} (black colored plot corresponding to the left axis) and σ_{xy} (orange colored plot corresponding to the right axis) vs. filling (ν) at constant magnetic fields for $s=-2$ (b) and $s=0$ (c) Chern gaps. The dashed horizontal lines correspond to $\sigma_{xy} = Ce^2/h$. The blue-shaded ν region (with a width of 0.2) is centered at the ν value calculated from the Diophantine equation for a particular (C, s) state at that B value. σ_{xx} shows a dip within the blue ν -window showing (C, s) Chern gaps. The value of σ_{xy} being close to Ce^2/h inside the blue-shaded region indicates approximate quantization.

The absence of a clear quantization in the Hall conductance from the Hofstadter butterfly can be understood by the low magnitude of the Hofstadter gaps. As revealed by our temperature-dependent transport, even the most prominent Hofstadter gaps have magnitudes of 0.1 to 0.4 meV (see Supplementary Fig. 2). Such low values of the gap can be due to the following reasons:

- Hofstadter physics generically involves gaps with low magnitude, because each Landau level in Hofstadter spectra actually splits into numerous subbands (fractal). For example, quantized QHSs were observed in graphene even at room temperature (Novoselov et al., *Science* **335**, 1379, 2007). In contrast, the observation of Hofstadter butterflies in graphene/hBN required a low temperature and high magnetic field (Hunt et al., *Science* **340**, 6139, 2013). Indeed, unlike Hofstadter physics, the physics of correlated Chern insulator states involve only a few bands: correlated Chern insulator states occur when the correlation effect separates otherwise degenerate four flat Chern bands.
- The low bandwidth of the underlying flat bands in twisted double bilayer graphene further minimizes the energy scale. We discussed this aspect in Supplementary Note 6 and Supplementary Fig. 11.
- The opposite values of Chern numbers from two valleys dictate opposite evolutions of the corresponding Hofstadter spectra. As discussed in Fig. 3 of the main manuscript, this can result in energy gaps in the K-valley spectrum filled by the energy levels of the K'-valley spectrum (and vice versa). Thus, the nontrivial topology of the zero-magnetic field band structure further obscures the Hofstadter gaps.

On top of the low energy gaps, the twist-angle inhomogeneity can further impede the observation of quantization. While angle disorder is a common problem in the twistrionics field, we would like to note that our devices are of state-of-the-art quality as reflected by the observation of correlated insulator gaps at zero magnetic field (see Supplementary Fig. 8a and 8b).

Finally, we note some interesting facts from the literature which somewhat contradict the reviewer's assessment about routine observation of quantization by other groups, especially in TDBG.

- The earlier experimental reports on twisted double bilayer graphene do not show a clear quantization in their magneto-transport data though they routinely assign Chern numbers solely by locating the minima in longitudinal resistance/conductance. For example, in Fig. R2, we put magneto-transport data from some novel reports that explore correlation physics in twisted double bilayer graphene. They do not show corresponding Hall conductance with clear quantization. Interestingly, often the same groups report quantization in the observation of correlated Chern insulator states in TBG. This corroborates that the energy scales play a greater role than the device quality in weakening the quantization.
- The first report on observing the giant anomalous Hall (AH) effect in TBG by Sharpe et al. *Science* **365**, 605 (2019) notes in their abstract, "the AH resistance is not quantized, and dissipation is present..." However, Serlin et al. *Science* **367**, 900 (2020) observed an "intrinsic quantized anomalous Hall effect" in the same system later.
- Despite the observation of quantized AH states in TBG, as noted in the last point, none of the two reports on AH states in TDBG shows quantization (He et al. arXiv:2109.08255, Kuiri et al. arXiv:2204.03442). In fact, arXiv:2109.08255, to which Reviewer 2 also referred, states: "The amplitude of the AHE is small in device O1, much less than the quantized value of h/e^2 ." While a very recent work by Liu et al. *Nat. Comm.* **13**, 3292 (2022) reports quantization, the quantization in TDBG devices seems elusive in general.

Fig. R2: Magneto-transport data from earlier published reports on TDBG. It illustrates that Chern numbers are routinely assigned by following the minima in longitudinal resistance without showing simultaneous quantization in corresponding transverse Hall conductance. Some of the minima, though labeled, are often obscured in the color-scale plots.

In essence, our observation of σ_{xx} minima and approximate quantization in σ_{xy} provide evidence of the (C, s) Chern states.

Action: In the revised Supplementary Information, we have now included the details of the Hall conductance data (Supplementary Fig. 5).

To sum it up, I think that the paper lacks sufficient novelty or a major advance. It also makes several questionable statements. Hence, I cannot recommend the manuscript for publication in Nature Communication.

Reply: We thank the reviewer for assessing our manuscript. However, we believe that our detailed response carefully addresses the possible issues. In summary, while small energy scales together with twist-angle inhomogeneity can hinder robust quantization in complex Hofstadter fractals in TDBG, our independent theoretical calculations further validate the experimental observation. In literature, the initial reports observing novel quantum states often lack robust quantization, however, they open up new directions. Similarly, we believe that our work will initiate further exploration of electric-field tuning of Chern numbers using Hofstadter physics.

REVIEWERS' COMMENTS

Reviewer #4 (Remarks to the Author):

I appreciate the effort of the authors to address my comments. However, I am afraid that my concerns remain the same. Most of the features that the authors claim to be novel in the introduction (e.g., opening a gap with an electric field) are not new. The original finding of the paper – the observed evolution of the Chern states with the perpendicular electric field and carrier density is curious but is not of high significance to the field.

1. Regarding the author's response to my comment #2: even after modification, I am afraid I still have to disagree with their statement "Recently a pure electrical control, such as the perpendicular electric field, to open up a Chern insulating state from a bulk gapless state has been demonstrated using a correlated system. Similar electrical control over Chern states without requiring electron correlation will be novel."

Opening up a Chern insulating state from a gapless state using an electric field is exactly what was demonstrated in the ten-year-old paper that I mentioned in my comment (PRL 108, 076601 (2012), Sanchez-Yamagishi et al.) and hence could hardly be considered novel. The displacement field allows one to tune between compressible states (when Landau levels cross) and incompressible states with finite Chern numbers, as is evident from Fig 2f in that paper. Moreover, in that case, the electronic correlations were indeed negligible in contrast to TDBG.

2. I want to discuss this last point further. Despite the authors repeatedly emphasize that their observations are unique because, as they claim, they do not require strong electronic correlations. The problem with this is that the systems that they study are still inherently strongly-correlated. For example, Cao et al. Nature 2020, observed correlated states in devices with nearly the same twist angle 1.1 deg. Moreover, the absence of a clear insulating state at half-fillings is not evidence of the absence of correlations. It is common to observe interaction-driven metallic states with reduced spin and valley degeneracy that show up as a "halo" in R_{xx} even without robust insulating states. For example, Shen et al. Nat Phys. 2020, observed this in 1.06deg TDBG devices. Authors themselves commented that they see such halo regions. Hence, it is very likely that the correlations in their devices are, in fact, strong and they will considerably affect the band structure and phenomenology of the system. I also want to note that the authors themselves invoke strong correlations in flat bands to explain strong spin splitting in their model (line 133). In such case, speculations that the observed physics doesn't require correlations seems to me unnecessary and even misleading.

3. In their manuscript and in the replies to my and other reviewer's comments, the authors often imply that the transition between the Chern numbers in their study is done with "only"

perpendicular electric field. I completely agree with the point that Reviewer #2 raised earlier, that in the end, to switch between the Chern states it is still required to change both displacement field and carrier density. The required change of the carrier density is exactly the same for a given field as that required for any other Landau levels. So from this standpoint, there is nothing new compared to the aforementioned TBG system (Sanchez-Yamagishi et al.) or any other Landau levels; only there one does not even need to change the displacement field to change the Chern number. The authors made a point that in their system, the gap changes Chern number at the same energy. However, i) they establish it only from simulation and not experiment; ii) this energy consideration does not seem to be of any particular significance since in vdW systems, the carrier density is the quantity that is tuned.

4. In the third paragraph of the manuscript, the authors write “Thus TDBG is a unique platform as the electric field plays an important role, unlike in TBG.” It is again hard to agree with this, because, in fact, besides TBG, all graphene moiré heterostructures have more than two layers and show displacement-field tunability of the band structure and Chern numbers. Examples: ABC trilayer, twisted mono-bi, alternating twist trilayer, and others. So TDBG is not unique at all.

5. Arguably, the central result of the paper is the data in Fig 2a and the simulations in Fig. 3d. The authors make the point that they match quite well. On the contrary, I see significant discrepancies. States (2,-2) and (0, 0) are experimentally observed at zero displacement field but only at finite field in simulations.

6. In the last sentence of the main text the authors write: “It is interesting to speculate that ferroelectric

correlations, as seen in recent experiments, could stabilize Chern bands and the physics we discuss in this study even at zero magnetic field. “

The paper discusses Hofstadter states which are fundamentally generated by magnetic fields and vanish without it. Furthermore, the ferroelectric polarization alone can not induce broken time-reversal symmetry, which is needed for a zero-field Chern state. Hence this speculation introduced at the end appears to be not substantiated and out of place.

7. A minor but important point: throughout the paper, the authors use the term “perpendicular electric field” but in the title they use a more ambiguous term “vertical electric field”.

We thank the reviewer for reviewing our manuscript. This helps us to improve our manuscript. Below, we respond to the reviewer's comments/questions:

(Reviewer's comments/questions are in *blue colored italic font* and our responses are in black colored font.)

Reviewer #4 (Remarks to the Author):

I appreciate the effort of the authors to address my comments. However, I am afraid that my concerns remain the same. Most of the features that the authors claim to be novel in the introduction (e.g., opening a gap with an electric field) are not new. The original finding of the paper – the observed evolution of the Chern states with the perpendicular electric field and carrier density is curious but is not of high significance to the field.

Reply: We appreciate the reviewer's comment. However, we strongly believe that the ability to switch Chern states by the perpendicular electric field in twisted double bilayer graphene (TDBG) is novel – it opens up a new direction for realizing magnetoelectric coupling. Furthermore, as Reviewer 3 earlier put it nicely, revealing the influence of band topology on Hofstadter Physics in our study is interesting and important in the context of current studies of twisted moiré systems. Additionally, reviewer 1 asserts that our work is novel and appropriate for publication in Nature Communications.

1. Regarding the author's response to my comment #2: even after modification, I am afraid I still have to disagree with their statement "Recently a pure electrical control, such as the perpendicular electric field, to open up a Chern insulating state from a bulk gapless state has been demonstrated using a correlated system. Similar electrical control over Chern states without requiring electron correlation will be novel."

Opening up a Chern insulating state from a gapless state using an electric field is exactly what was demonstrated in the ten-year-old paper that I mentioned in my comment (PRL 108, 076601 (2012), Sanchez-Yamagishi et al.) and hence could hardly be considered novel. The displacement field allows one to tune between compressible states (when Landau levels cross) and incompressible states with finite Chern numbers, as is evident from Fig 2f in that paper. Moreover, in that case, the electronic correlations were indeed negligible in contrast to TDBG.

Reply: We thank the reviewer for the comment. We would like to reiterate that our demonstration of electrical control over Chern states includes but is not limited to tuning between compressible states and incompressible states. Together with the electric field tunable band structure and a plethora of Chern states in the Hofstadter spectra, TDBG allows switching among different Chern states. In other words, while tuning from $C = 0$ to $C = \text{nonzero}$ state has been explored earlier, on-demand switching between different values of finite C using the electric field, as demonstrated in our work, is novel.

2. I want to discuss this last point further. Despite the authors repeatedly emphasize that their observations are unique because, as they claim, they do not require strong electronic correlations. The problem with this is that the systems that they study are still inherently strongly-correlated. For example, Cao et al. Nature 2020, observed correlated states in devices with nearly the same twist angle 1.1 deg. Moreover, the absence of a clear insulating state at half-fillings is not evidence of the absence of correlations. It is common to observe interaction-driven metallic states with reduced spin and valley degeneracy that show up as a “halo” in R_{xx} even without robust insulating states. For example, Shen et al. Nat Phys. 2020, observed this in 1.06deg TDBG devices. Authors themselves commented that they see such halo regions. Hence, it is very likely that the correlations in their devices are, in fact, strong and they will considerably affect the band structure and phenomenology of the system. I also want to note that the authors themselves invoke strong correlations in flat bands to explain strong spin splitting in their model (line 133). In such case, speculations that the observed physics doesn't require correlations seems to me unnecessary and even misleading.

Reply: We thank the reviewer for the comment. However, we would like to clarify our point that the physics of electric field tunable Chern states using Hofstadter spectra of TDBG does not require electronic correlations as a *prerequisite*. This is evident from two facts:

- (i) Our experimental results are consistent across multiple devices with twist angles 1.09°, 1.10°, 1.42°, and 1.46° (e.g., see Supplementary Figures 7-9). Different twist angles provide different degrees of correlation.
- (ii) Our independent theoretical calculations confirm the basic observation (i.e., the cascade of Chern transitions by varying the electric field) without invoking the electron correlation effect.

Of course, the finer details such as strong spin splitting depend on the presence of correlations. In other words, the correlation effect is not a prerequisite for the physics we observed but may provide richer interplay.

Action: To make the message clearer, we have modified the sentence under the Discussion section ‘Our work demonstrates a novel pathway to control Chern states using the electric field without requiring electron correlation as a prerequisite.’

3. In their manuscript and in the replies to my and other reviewer's comments, the authors often imply that the transition between the Chern numbers in their study is done with “only” perpendicular electric field. I completely agree with the point that Reviewer #2 raised earlier, that in the end, to switch between the Chern states it is still required to change both displacement field and carrier density. The required change of the carrier density is exactly the same for a given field as that required for any other Landau levels. So from this standpoint, there is nothing new compared to the aforementioned TBG system (Sanchez-Yamagishi et al.) or any other Landau levels; only there one does not even need to change the displacement field to change the Chern number. The authors made a point that in their system, the gap changes Chern number at the same energy. However, i) they establish it only from simulation and not experiment; ii) this energy consideration does not seem to be of any particular significance since in vdW systems, the carrier density is the quantity that is tuned.

Reply: We thank the reviewer for the comment. It is common for theory and experiment to work synergistically to provide insights into a physical system. Our independent theoretical calculations support our experimental findings and provide an understanding that experiments alone cannot provide. The physics of layer polarization is very elegantly revealed from our theory calculation.

4. In the third paragraph of the manuscript, the authors write “Thus TDBG is a unique platform as the electric field plays an important role, unlike in TBG.” It is again hard to agree with this, because, in fact, besides TBG, all graphene moiré heterostructures have more than two layers and show displacement-field tunability of the band structure and Chern numbers. Examples: ABC trilayer, twisted mono-bi, alternating twist trilayer, and others. So TDBG is not unique at all.

Reply: We thank the reviewer for the comment. Our comment is in the context of twisted systems. In the earlier rounds, reviewers had asked us to contrast our findings with twisted bilayer graphene as it has received significantly more attention – this is the genesis of this statement. It goes without saying that the electric field will play an important role for many more layers of graphene as the potential on the extreme layers will be larger.

Action: To remove any potential confusion we have modified the statement in the revised manuscript as, “Thus, TDBG is a rich platform as the electric field plays an important role, unlike in TBG.”

5. Arguably, the central result of the paper is the data in Fig 2a and the simulations in Fig. 3d. The authors make the point that they match quite well. On the contrary, I see significant discrepancies. States (2,-2) and (0, 0) are experimentally observed at zero displacement field but only at finite field in simulations.

Reply: We appreciate the comment of the reviewer. We first note that while we have used the most common form of TDBG Hamiltonian, different parameters of the Hamiltonian are still undetermined from experiments and vary across theoretical reports. Despite this, though there are some quantitative differences, our theoretical simulation captures the Chern bands with correct topological indices qualitatively and helps us characterize the nature of the transition from one gap to another.

Secondly, the disagreement in the electric field axis is not fundamental and can be attributed in a realistic system to the extra potential on each layer induced by the presence of other layers. So, the effective potential on each layer could be different from the applied potential. In principle, a self-consistent calculation could capture the actual potential present in the layers. Unfortunately, such computation is numerically quite expensive, whereas the resulting solutions will simply differ by certain shifts of potential. In our calculation, we kept the potentials on each layer fixed ($3/2V$, $1/2V$, $-1/2V$, $-3/2V$), instead of determining them self-consistently.

Finally, we would like to add that calculating Hofstadter spectra in literature are mostly limited to simple model Hamiltonians, e.g., square potential. The ability to incorporate richer Hamiltonian

such as TDBG helps us unveil the physics of layer polarization. More detailed calculations in the future can offer further quantitative details.

Action: We have included the possible role of additional potential difference induced due to the presence of other layers in the revised Supplementary Note 2.

6. In the last sentence of the main text the authors write: "It is interesting to speculate that ferroelectric correlations, as seen in recent experiments, could stabilize Chern bands and the physics we discuss in this study even at zero magnetic field. "The paper discusses Hofstadter states which are fundamentally generated by magnetic fields and vanish without it. Furthermore, the ferroelectric polarization alone can not induce broken time-reversal symmetry, which is needed for a zero-field Chern state. Hence this speculation introduced at the end appears to be not substantiated and out of place.

Reply: We thank the reviewer for the comment. While we agree that ferroelectric polarization alone is insufficient, together with the underlying correlations they provide rich opportunities. Here we would like to note that it is common to delineate the implications of the work for the readers toward the end of the manuscript. Often these implications are speculative in nature. However, they can provide new future directions.

7. A minor but important point: throughout the paper, the authors use the term "perpendicular electric field" but in the title they use a more ambiguous term "vertical electric field".

Reply: We thank the reviewer for this point.

Action: We have changed the terminology in the title to make it consistent with the rest of the paper.